# Mechanism of replication origin melting nucleated by CMG helicase assembly

Jacob S. Lewis[1], Marta H. Gross[2,5], Joana Sousa[1,4,5], Sarah S. Henrikus[1], Julia F. Greiwe[1], Andrea Nans[3], John F. X. Diffley[2] & Alessandro Costa[1✉]

The activation of eukaryotic origins of replication occurs in temporally separated steps to ensure that chromosomes are copied only once per cell cycle. First, the MCM helicase is loaded onto duplex DNA as an inactive double hexamer. Activation occurs after the recruitment of a set of firing factors that assemble two Cdc45–MCM–GINS (CMG) holo-helicases. CMG formation leads to the underwinding of DNA on the path to the establishment of the replication fork, but whether DNA becomes melted at this stage is unknown[1]. Here we use cryo-electron microscopy to image ATP-dependent CMG assembly on a chromatinized origin, reconstituted in vitro with purified yeast proteins. We find that CMG formation disrupts the double hexamer interface and thereby exposes duplex DNA in between the two CMGs. The two helicases remain tethered, which gives rise to a splayed dimer, with implications for origin activation and replisome integrity. Inside each MCM ring, the double helix becomes untwisted and base pairing is broken. This comes as the result of ATP-triggered conformational changes in MCM that involve DNA stretching and protein-mediated stabilization of three orphan bases. Mcm2 pore-loop residues that engage DNA in our structure are dispensable for double hexamer loading and CMG formation, but are essential to untwist the DNA and promote replication. Our results explain how ATP binding nucleates origin DNA melting by the CMG and maintains replisome stability at initiation.

Since the discovery of the double helix, molecular biologists have been asking how the separation of two DNA strands is nucleated after the initiation of chromosome replication. In vitro reconstitution of bacterial[2], viral[3] and eukaryotic DNA replication[1] have started to address this question. By studying these systems, a universal role has been identified for ATP binding by multimeric enzymes that untwist the double helix at the start of replication. Existing atomic models of initiators and helicases bound to single-stranded DNA mimic the structure of origin DNA immediately after melting[2,4,5]. However, to understand the mechanism of the ATP-triggered opening of duplex DNA at the molecular level, the structure of an origin duplex caught in the act of nucleating a replication bubble must be obtained. To achieve the opening of origin DNA, bacteria[2] and eukaryotic viruses[3,6] use one single protein that oligomerizes around the double helix and causes its deformation, but such melting intermediates have not to our knowledge been structurally characterized so far. Origin opening in *Saccharomyces cerevisiae* requires not one, but thirty-two distinct polypeptides that act sequentially. First, the origin recognition complex (ORC) together with loading factors Cdc6 and Cdt1 recruit a set of two ring-shaped MCM helicases that form an inactive double hexamer around duplex DNA[7,8]. Activation requires the recruitment of two firing factors, Cdc45 and Go-Ichi-Nii-San (GINS)[1,9–11]. To achieve this, the double hexamer is first phosphorylated by the Dbf4-dependent kinase (DDK). These changes are recognized by the Sld3–7 phosphoreader, which recruits

Cdc45 to the double hexamer[11–15]. Sld3 is in turn phosphorylated by the Clb5–Cdc28 (CDK) kinase, which also phosphorylates the firing factor Sld2. Phospho-Sld2 and phospho-Sld3 bind Dpb11, which engages Pol ε and GINS to mediate their origin recruitment[11,12,16]. After ADP release and ATP binding by MCM, GINS and Cdc45 stably engage MCM, forming two distinct CMG assemblies that disrupt the double hexamer interface through an unknown mechanism. Topology footprint assays indicate that CMG formation leads to partial DNA untwisting, but whether base pairing is broken at this stage in origin activation remains to be determined. After the recruitment of Mcm10, the lagging-strand template is ejected from the MCM ring pore, which leads to the establishment of the replication fork and the ATPase-powered translocation along single-stranded DNA[1]. How the CMG selects the translocation strand in this context is unknown. Assembly of two CMGs at an origin disrupts the double hexamer interface[1]. Mapping the relative orientation of the two separated CMGs on the origin DNA is important to understand how replication forks are established bidirectionally and how replisome stability is maintained in the early stages of replication initiation[17–22].

## CMG assembly on chromatinized origin DNA

To understand how ATP-dependent CMG formation leads to double hexamer separation and DNA untwisting, we used electron microscopy to image origin-dependent CMG formation reconstituted in vitro with

[1]Macromolecular Machines Laboratory, The Francis Crick Institute, London, UK. [2]Chromosome Replication Laboratory, The Francis Crick Institute, London, UK. [3]Structural Biology Science Technology Platform, The Francis Crick Institute, London, UK. [4]Present address: UCB Pharma, Slough, UK. [5]These authors contributed equally: Marta H. Gross, Joana Sousa. ✉e-mail: alessandro.costa@crick.ac.uk

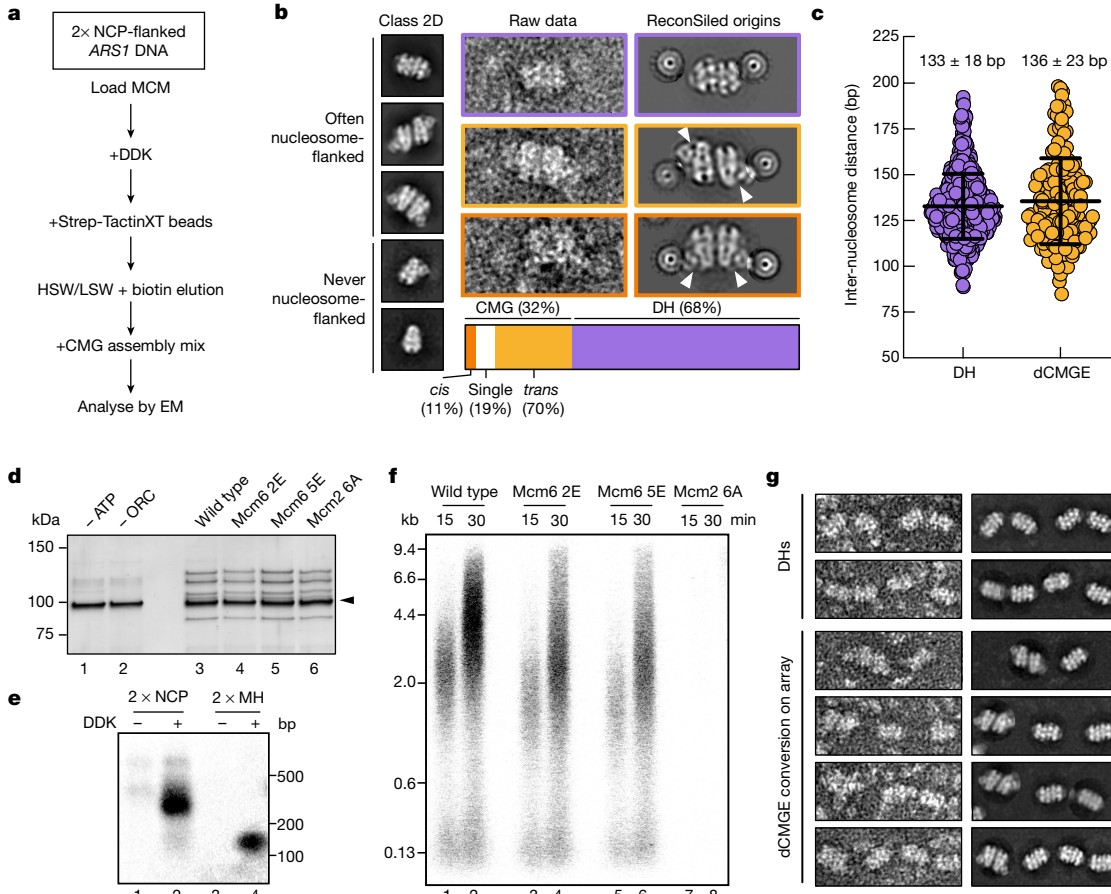

**Fig. 1 | Visualization of origin-dependent CMG assembly by electron microscopy. a**, Workflow for the assembly of CMG on a chromatinized origin of replication for electron microscopy (EM) imaging. HSW, high-salt wash; LSW, low-salt wash; NCP, nucleosome core particle. **b**, Left, 2D averages derived from NS-EM imaging of the CMG assembly reaction. Centre, raw images and right, in silico reconstitution (ReconSil) of the double hexamer (DH) or dCMGE particles on the chromatinized origin of replication. Bottom, representation of the double-hexamer-to-CMG conversion efficiency. **c**, Measure of inter-nucleosome distance matches the expected length of the *ARS1* origin of replication (*n* = 444 origins for double hexamer; *n* = 186 origins for dCMGE). Error bars, mean ± s.d. **d**, Comparison between MCM loading on short DNA containing MH roadblocks. After HSW treatment, equal amounts of loaded MCM helicases are eluted from

Strep-TactinXT beads. The black arrowhead indicates MH-bound DNA. For gel source data, see Supplementary Fig. 1. This experiment was performed twice. **e**, Analysis of the replication products by alkaline agarose gel electrophoresis indicates that short nucleosome- and MH-capped origins can be replicated. For gel source data, see Supplementary Fig. 1. This experiment was performed twice. **f**, Replication reaction performed as shown in **d** except on large *ARS1* circular DNA of wild-type and mutant MCMs. Mutants include Mcm2 6A, which targets residues that are involved in DNA untwisting; Mcm6 2E, which targets the Mcm6 wedge insertion; and Mcm6 5E, which targets the safety latch. For gel source data, see Supplementary Fig. 1. This experiment was performed twice. **g**, ReconSil of dCMGE formation on a 6× *ARS1* array built from loaded double hexamers. This experiment was performed three times.

purified yeast proteins and in a near-native chromatin environment. To this end, we reconstituted CMG on *ARS1* origin DNA flanked at both ends by a nucleosome assembled on strong Widom positioning sequences. Nucleosome capping of the naked, AT-rich *ARS1* DNA recapitulates the architecture of chromatinized origins that is found in cells[23] and serves to trap double hexamers on duplex DNA, preventing dissociation by sliding[24]. Double hexamers were (i) loaded onto origin DNA using MCM–Cdt1, ORC, Cdc6 and ATP; (ii) phosphorylated with DDK in solution; and (iii) isolated using Strep-TactinXT-coated paramagnetic beads that capture a twin-strep tag on histone H3 of the nucleosome. After a high-salt wash that removes helicase loading intermediates and DDK, DNA-bound phosphorylated double hexamers were biotin-eluted and incubated with Sld3–7, Cdc45, Sld2, Dpb11, GINS, Pol ε, CDK and ATP to promote CMG formation (Fig. 1a and Extended Data Fig. 1a,b). We analysed the full reaction by negative-stain electron microscopy (NS-EM) single-particle two-dimensional (2D) averaging, to find that on average 32% of double hexamers were converted to CMG. Of these, 70% were homo-dimeric and Pol-ε-engaged (dCMGE) and 19% were single CMGs. In most of the dCMGE particles, GINS–Cdc45 and Pol ε mapped

on opposite sides around the MCM ring, giving rise to a *trans* configuration (Fig. 1b and Extended Data Fig. 1c). The remaining dCMGE particles (11%) were in a *cis* configuration, with GINS–Cdc45–Pol ε located on the same side (Fig. 1b and Extended Data Fig. 1c). In silico reconstitution (ReconSil), performed by overlaying nucleosome and MCM-containing 2D averages onto the corresponding particles in the raw micrographs, revealed that dCMGEs were tightly packed between the two flanking nucleosomes (Fig. 1b). A measured inter-nucleosome distance of 136 bp ± 23 bp (s.d.) (Fig. 1c) matches the expected 136 bp, supporting the notion that these represent bona fide reconstituted origins and not neighbouring particles bound to different DNA molecules. Similar results were obtained for chromatinized origins trapping a double hexamer (133 bp ± 18 bp (s.d.); Fig. 1c). We did not observe single CMGs trapped in between nucleosomes, suggesting that these may represent helicases that fell off the DNA (Fig. 1b). It is established that *ARS1* DNA substrates capped by nucleosomes, or by covalently linked HpaII methyltransferase (MH) roadblocks, support double hexamer loading[24] (Fig. 1b,d). Given the tighter protein packing that is caused by the formation of dCMGEs on the *ARS1* origin, we asked whether

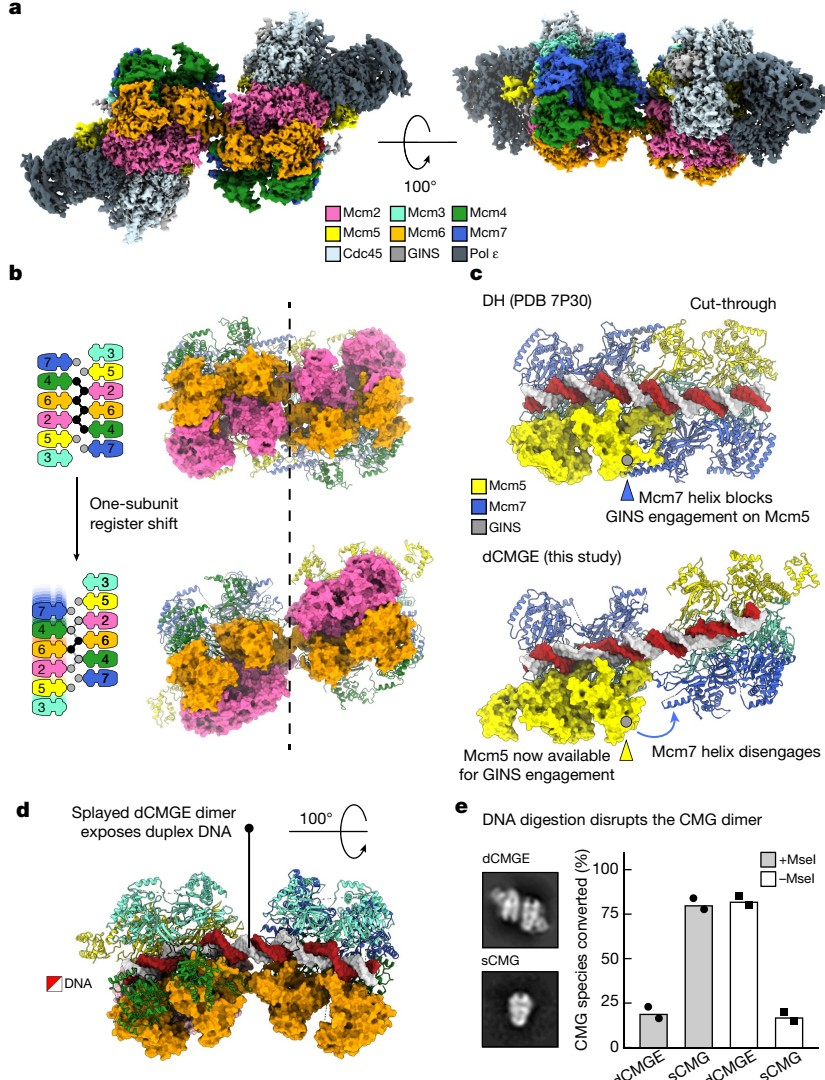

**a**

Mcm2 | Mcm3 | Mcm4
Mcm5 | Mcm6 | Mcm7
Cdc45 | GINS | Pol ε

**b** One-subunit register shift

**c** DH (PDB 7P30) | Cut-through

Mcm5
Mcm7
GINS

Mcm7 helix blocks GINS engagement on Mcm5

dCMGE (this study)

Mcm5 now available for GINS engagement | Mcm7 helix disengages

**d** Splayed dCMGE dimer exposes duplex DNA

100°

DNA

**e** DNA digestion disrupts the CMG dimer

dCMGE

sCMG

CMG species converted (%)

+MseI | −MseI

dCMGE | sCMG | dCMGE | sCMG

**Fig. 2 | dCMGE formation reconfigures the double hexamer interface, resulting in a splayed dimer. a**, Surface rendering of the dCMGE complex. **b**, Double-hexamer-to-dCMGE conversion promotes a one-subunit register shift at the MCM dimerization interface. Circles represent ZnFs. Black circles connected by lines indicate ZnFs engaged in tight inter-ring interactions. **c**, Double-hexamer-to-dCMGE conversion promotes the disengagement of an Mcm7 α-helical extension that protects the Mcm5 A domain on the opposite ring. This structural change exposes a GINS-binding site on Mcm5. PDB 7P30 refers to the Protein Data Bank (PDB) accession code. **d**, The dCMGE dimer is held together by a Mcm6 homo-dimer as well as by the DNA duplex. The dCMGE splayed dimer exposes a stretch of twisted duplex DNA that intervenes between the two MCM rings. **e**, DNA digestion disrupts the dCMGE dimer into single isolated CMGs (sCMGs), while also promoting the disengagement of Pol ε. This experiment was performed twice. Mean values are shown.

enough space is available between nucleosomes for two activated CMGE particles to cross paths during the establishment of the replication fork. To address this question, we reconstituted roadblocked origin replication in vitro using a minimal set of replisome factors (Fig. 1d,e), matching established conditions that support the replication of an ARS-containing 10.6-kb supercoiled plasmid[11] (Fig. 1f). DNA products separated by alkaline agarose gel electrophoresis showed that the nucleosome-flanked *ARS1* substrate is copied in full (Fig. 1e). This is evident from the size of duplicated DNA, which is longer for the Widom-flanked *ARS1*, compared to a shorter construct in which nucleosomes are swapped for MH caps (Extended Data Fig. 1b and Fig. 1e). The tight packing of a dCMGE particle between two flanking nucleosomes raises the question of whether the CMGE dimer is a stable complex or whether it is formed as a result of the spatial constraints imposed by the two roadblocks that prevent dissociation. We reasoned that CMG assembly on longer, less crowded DNA substrates might allow enough

space for the dCMGE complex to dissociate into two discrete CMGE particles. To test this hypothesis, we performed origin-dependent CMG assembly reactions on MH-capped 864-bp DNA substrates that contain an array of 6 consecutive *ARS1* sequences separated by 40 bp of linker DNA. We only observed dCMGE particles and not separated CMGs on the array substrate, irrespective of the efficiency of double hexamer loading (Fig. 1g and Extended Data Fig. 1d–g). Thus, stability of the dCMGE complex assembled during ATP-dependent double hexamer activation is independent of nucleosomes and independent of the position of flanking roadblocks, as well as the level of protein saturation of DNA.

## Cryo-EM structure of the CMGE dimer

Our previous NS-EM work on origin-dependent CMG reconstitution in vitro involved high-salt treatment of protein–DNA tethered to

streptavidin-coated paramagnetic beads, followed by elution using DNA digestion[1]. This procedure disrupted CMGE dimers but not the double hexamer, which suggests that the conversion of double hexamer to dCMGE reconfigures and weakens the MCM dimerization interface. To understand the conformational transitions that occur upon CMG formation, we determined the cryo-electron microscopy (cryo-EM) structure of a dCMGE complex assembled on chromatinized *ARS1* DNA (Extended Data Fig. 2a–c). Both *C*2 symmetry and asymmetric refinement yielded a structure with limited resolution (Extended Data Figs. 2d–f and 3). However, symmetry expansion approaches revealed that the two rings in the dimer are identical. Combined with three-dimensional (3D) classification, variability analysis and refinement, followed by iterative cycles of contrast transfer function (CTF) refinement and Bayesian polishing, this process yielded a structure at 3.5 Å resolution[25–27] (or 3.4 Å after density modification[28]; Fig. 2a, Supplementary Video 1, Extended Data Figs. 2g–j, 3 and 4 and Extended Data Table 1). By overlaying two copies of the CMGE monomer to the lower-resolution dimer, we can therefore obtain a high-resolution view of the entire dCMGE assembly. While in the double hexamer the tight homo-dimerization interface is formed by the packing between six MCM zinc-finger domains (ZnFs; degenerate in Mcm3), the transition to dCMGE involves the loss of several trans-ring interactions and a one-subunit register shift for the remaining tethering elements. The residual inter-ring contacts involve the Mcm6 ZnF that transitions from an Mcm6–Mcm6–2 to a Mcm6–Mcm4–6 interaction (Fig. 2b and Supplementary Video 2). What role this register shift might have during origin unwinding is addressed in the Supplementary Discussion. As the Mcm5–7 *trans* contact is disrupted after double-hexamer-to-dCMGE conversion, a helical insertion in the A domain of Mcm7 disengages from the Mcm5 A domain on the opposed ring, releasing a steric impediment that would prevent stable CMG assembly by hindering the association of GINS with Mcm5 (ref. [5]) (Fig. 2c and Supplementary Video 2). As a result, the two CMG particles in the dCMGE complex become splayed open, pivoting around Mcm6 and creating a large cavity that exposes 1.5 turns of duplex DNA. The intervening DNA appears to stabilize the dCMGE dimer interface (Fig. 2d). In fact, partial digestion with the restriction enzyme MseI promotes the disassembly of the dCMGE complex into separated CMGs (Fig. 2e and Extended Data Fig. 1g).

## Nucleotide and DNA binding in the dCMGE complex

Structural analysis of the double-hexamer-to-dCMGE transition indicates that three sites (Mcm2>5, 5>3 and 3>7) exchange ADP for ATP, whereas Mcm7>4 remains ADP-bound. Mcm4>6, which is nucleotide-free in the double hexamer, is ADP-bound in the dCMGE complex, suggesting consecutive ATP binding and hydrolysis on the path to dCMGE formation. The Mcm6>2 interface instead transitions from an ATP-bound state in which the bipartite catalytic site is open and hydrolysis-incompetent, to a more compact, hydrolysis-competent state that contains a mixture of ADP and ATP in the nucleotide-binding pocket (Fig. 3a,b and Supplementary Video 3). Overall, four of the six ATPase sites of MCM release ADP and bind ATP, in agreement with an analysis of bound nucleotides by thin-layer chromatography[1]. This nucleotide exchange in turn alters the way in which duplex DNA is gripped by ATPase pore loops of the MCM. Although in the double hexamer, MCM engages in the same number of contacts with both the leading and the lagging strands[29,30], the vast majority of MCM interactions in the dCMGE involve the leading-strand template (Extended Data Fig. 5a). The ATPase domains of Mcm3, Mcm5, Mcm2 and Mcm6 provide most of the leading-strand contacts, which explains how the double-hexamer-to-dCMGE transition leads to selection of the translocation strand (Fig. 3c). This leading-strand binding mode in dCMGE is consistent with previously reported structures of single- and duplex-DNA-engaged recombinant CMG (refs. [17,31–34]), which indicates

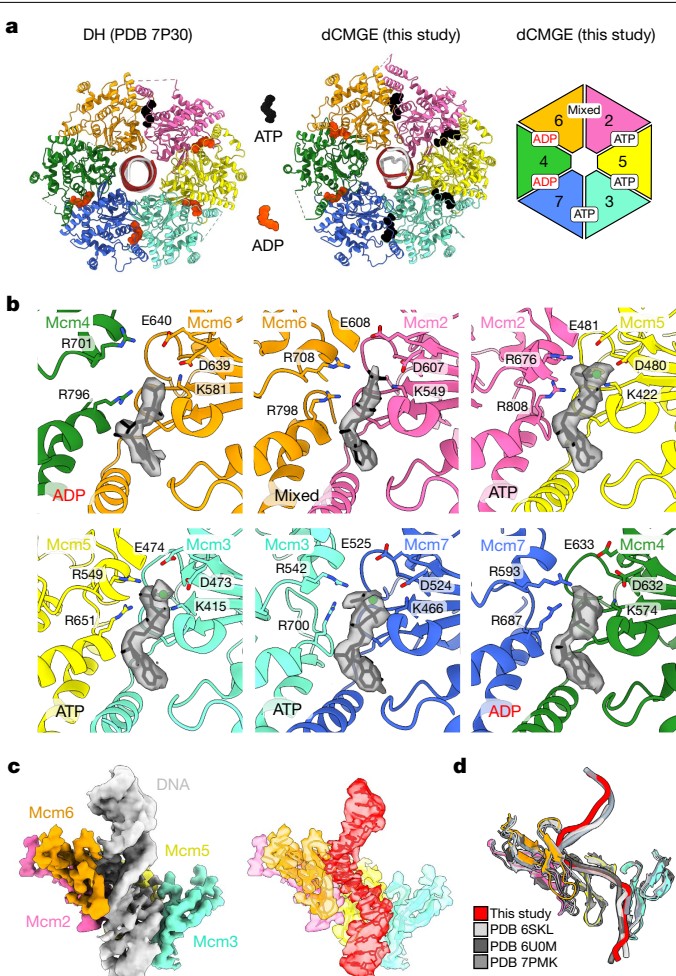

**Fig. 3 | ATP and DNA binding in the dCMGE complex. a**, MCM nucleotide occupancy in the double hexamer and in the dCMGE complex. **b**, Surface rendering of the nucleotide in the six ATPase sites of MCM. **c**, Duplex DNA binding in the dCMGE complex (left) explains how the double-hexamer-to-dCMGE transition leads to selection of the translocation strand. The ATPase pore loops in the dCMGE complex only contact the leading-strand template. The density for the selected translocation strand (red on the right) has been extracted from the duplex DNA density (grey on the left). **d**, The leading-strand template extracted from the dCMGE structure superposed on the yeast CMG translocating on a DNA fork reconstituted on an artificial DNA fork (PDB 6U0M), bound to the fork stabilization complex (PDB 6SKL) or bound to SCF^Dia2 and duplex DNA (PDB 7PMK).

that interaction with the leading strand is conserved from initiation to termination (Fig. 3d). Given the sparse pore-loop contacts, the lagging-strand template appears to be in turn poised for ejection from the MCM ring pore, which is required to achieve the establishment of the replication fork. How nucleotide-triggered conformational changes in MCM affect the structure of duplex DNA will be discussed in the next paragraph.

## Mechanism of DNA-bubble nucleation

Captured between the ATPase and the N-terminal pore loops in the dCMGE complex, a stretch of seven base pairs is underwound. This observation agrees with previous topology footprint measurements that indicate that 0.7 turns of DNA become untwisted per CMG complex, after the double-hexamer-to-CMG transition in the absence of Mcm10 (ref. [1]). On the basis of the cryo-EM density of the untwisted

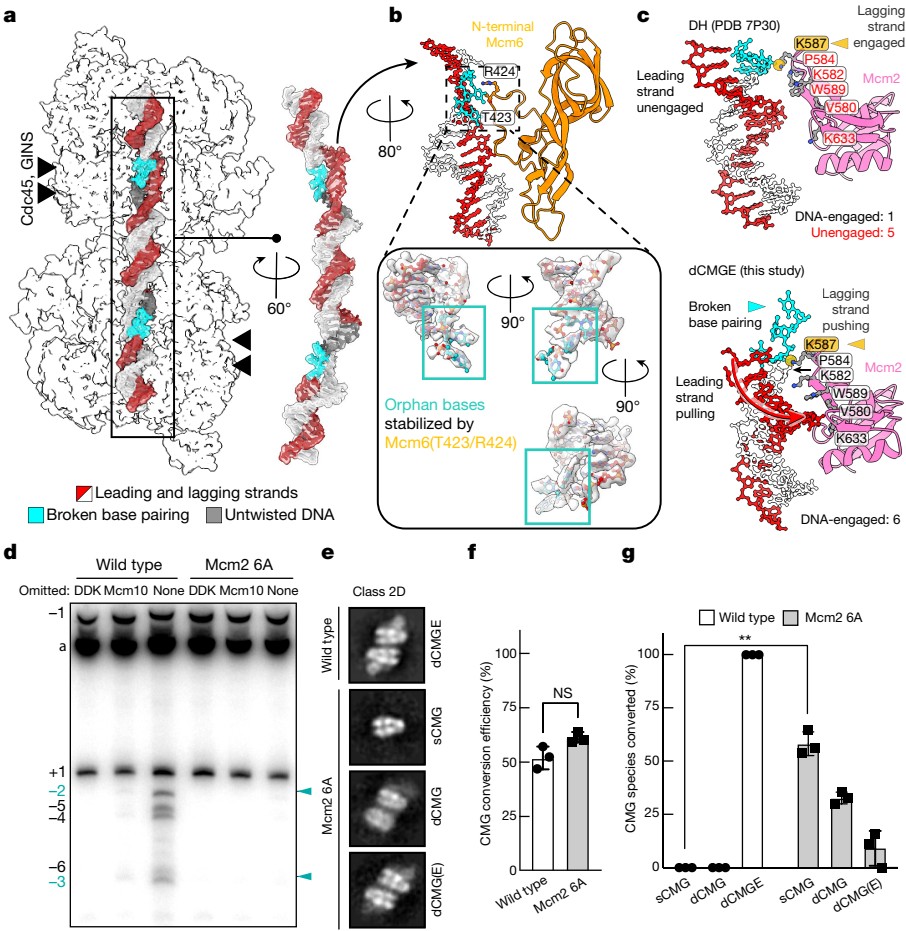

**Fig. 4 | dCMGE formation leads to the untwisting of duplex DNA and breaks at least three base pairs. a**, Cryo-EM density of origin DNA. **b**, 0.7 turns of the double helix become untwisted after dCMGE formation. Three orphan bases become stabilized by residues T423 and R424 from the Mcm6-specific N-terminal hairpin insertion. **c**, Within the Mcm2 ATPase domain, only residue K587 contacts duplex DNA in the double hexamer. In dCMGE, five additional Mcm2 ATPase residues contact the DNA, which promotes widening of the minor groove, untwisting of duplex DNA and disruption of base pairing. **d**, Topology footprint assay for DNA unwinding. Complete reactions contained all firing factors after MCM loading plus TopoI; omission of DDK blocks all untwisting. Omission of Mcm10 captures the initial untwisted state[1]. This initially untwisted state generates topoisomers of −2 and −3 (cyan arrowheads) as previously observed[1]. Additional negatively supercoiled topoisomers can be detected when Mcm10 is present, indicating further untwisting after ejection of the lagging strand from CMG. No topoisomers were observed with the

Mcm2 6A mutant. For gel source data, see Supplementary Fig. 1. This experiment was performed twice. **e**, NS-EM CMG averages derived from the CMG assembly reaction using wild-type or Mcm2 6A MCMs. CMG assembly reactions were performed on a 2 × MH DNA template (Extended Data Fig. 1c) to reduce background single CMG particles that are present owing to incomplete roadblocking of DNA. dCMGE is the product of CMG assembly using wild-type proteins, whereas Mcm2 6A primarily forms single CMGs. A minority of particles are compatible with dCMG or dCMGE formation (although in the latter, Pol ε occupancy is only partial (dCMG(E))). **f**, Mcm2 6A MCMs are converted to CMG at wild-type levels. *P* values were determined by two-tailed Welch's *t*-test; NS, not significant. This experiment was performed three times. Error bars, mean ± s.d. **g**, Mcm2 6A disrupts the dCMGE dimer mostly into single isolated CMGs. *P* values were determined by two-tailed Welch's *t*-test; **P = 0.0030. This experiment was performed three times. Error bars, mean ± s.d.

lagging strand, three flipped-out bases can be confidently built (Fig. 4a), which are stabilized by two conserved residues (T423 and R424) located within the Mcm6-specific insertion of the N-terminal pore loop ('Mcm6 wedge'; Fig. 4b and Extended Data Fig. 5b,c,e). Together, our data show that the double-hexamer-to-dCMGE transition not only promotes the untwisting of duplex DNA but also the disruption of at least three consecutive base pairs, with the resulting orphan bases being stabilized by an Mcm6 pore-loop element. Two separate bubbles are nucleated inside the two MCM rings across the dCMGE, which remain separated by 1.5 turns of exposed duplex DNA.

Structural changes in the ATPase pore loops explain how the double hexamer-to-dCMGE transition leads to DNA untwisting. Amongst several MCM–DNA interactions that are summarized in Extended Data Fig. 5a–d, we identified K587 on the Mcm2 helix-2 insert (h2i) pore loop as one of a few elements that maintain the same DNA contact in both

the double hexamer and the dCMGE complex. As shown in Supplementary Video 4, this element appears to push on the lagging-strand template, contributing to the deformation of duplex DNA. Additional DNA contacts involve five conserved residues on Mcm2 that pull on the leading-strand template (V580, K582, P584 and W589 in h2i, as well as K633 in the pre-sensor 1 β-hairpin, PS1BH; Extended Data Figs. 5f and 6a). These contacts are found in the dCMGE complex but are absent in the double hexamer and widen the minor groove, which decreases the superhelical DNA density (Fig. 4c). An Mcm2 variant (Mcm2 6A), which targets all of the ATPase–DNA contacts that are observed in the dCMGE complex, can load double hexamers to wild-type levels but is completely defective for replication (Fig. 1d,f). To establish whether this defect is due to the inability of Mcm2 6A to untwist DNA upon CMG assembly, we loaded double hexamers on a 616-bp circular DNA that contains *ARS1* and added a full complement of wild-type firing factors,

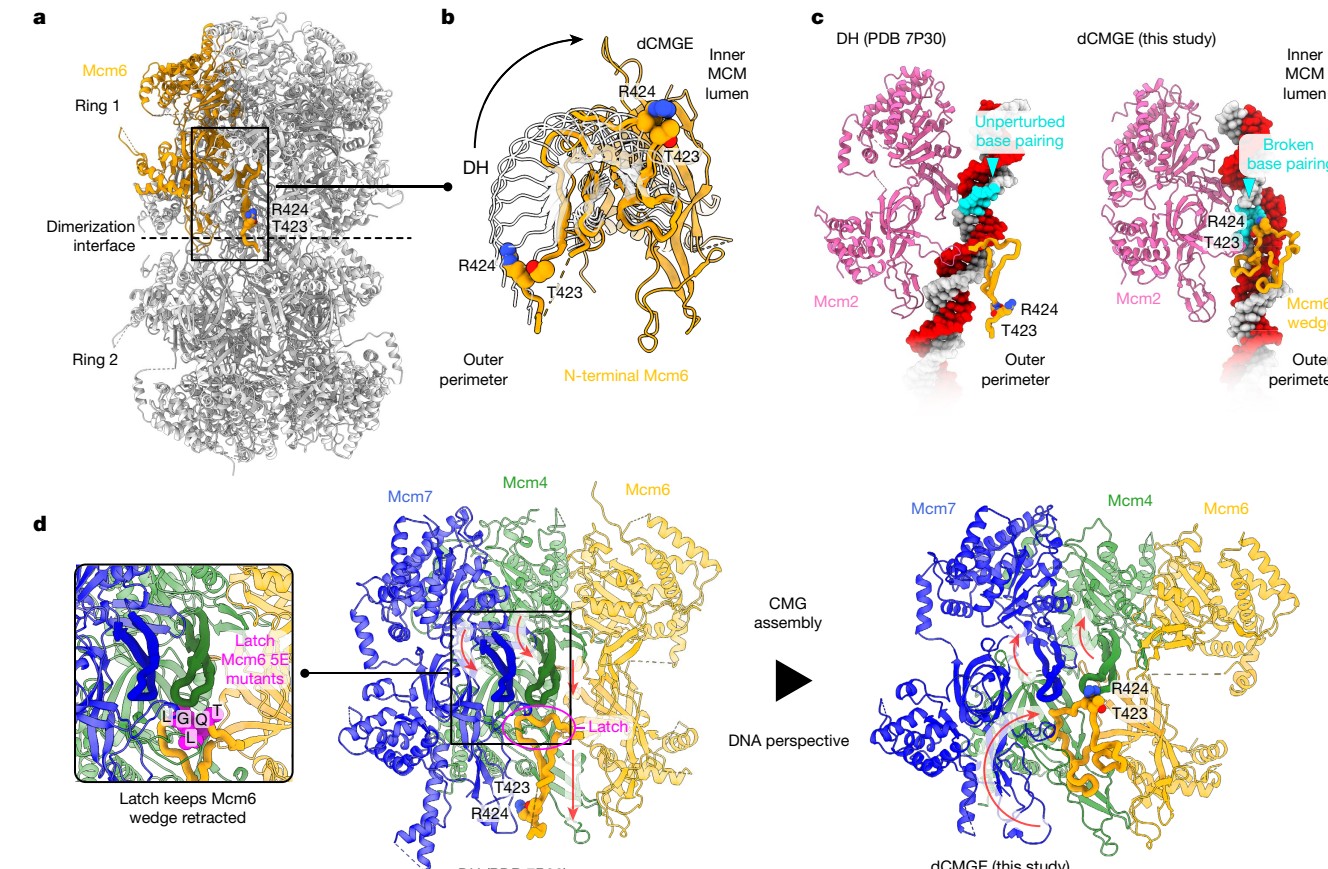

**Fig. 5 | A change in nucleotide engagement promotes the coupled disruption of the double hexamer dimerization interface and the stabilization of three orphan bases in the origin DNA duplex. a**, The Mcm6-specific wedge insertion in the N-terminal β-hairpin forms part of the dimerization interface in the double hexamer. In this configuration, wedge residues T423 and R424 map on the outer surface of the double hexamer. **b**, The double-hexamer-to-dCMGE transition promotes a reconfiguration of the Mcm6 wedge insertion, with T423 and R424 transitioning from the outer MCM perimeter to the inner lumen of the MCM ring. **c**, Swinging of the Mcm6 wedge from the outer MCM surface to the inner lumen leads to the stabilization of three orphan bases in the untwisted origin DNA duplex. **d**, In the double hexamer, the Mcm4 and Mcm7 h2i pore loops face downwards, with Mcm4 pushing against the Mcm6 N-terminal β-hairpin. This functions as a latch that maintains the Mcm6 wedge packed against the double hexamer dimerization interface. After CMG assembly, global changes in the ATPase tier of MCM cause the Mcm4 and Mcm7 h2i pore loops to move upwards, which releases the safety latch of the Mcm6 wedge insertion. This change promotes the Mcm6 wedge to swing upwards, with the R423 and T424 elements entering the MCM lumen to stabilize three orphan bases.

or selected dropout controls, in the presence of TopoI (Fig. 4d and Extended Data Fig. 6b,c). As previously described, three topoisomers, α, α−1 and α+1, were visible in the absence of DDK, indicating that there was no MCM-dependent change in DNA topology[1]. Omission of Mcm10 with wild-type MCM led to an additional accumulation of negatively supercoiled topoisomers −2 and −3, which are indicative of initial DNA untwisting[1] consistent with that seen in the dCMGE structure. When all firing factors were present with wild-type MCM, a robust accumulation of −2 through to −6 negative supercoils was detected (lane 3), indicating additional DNA untwisting that arises after the ejection of the lagging strand from CMG. None of the topoisomers that are associated with either initial untwisting or full activation appeared when MCM containing the Mcm2 6A variant was assayed. This indicates that DNA engagement by Mcm2—as observed in our dCMGE structure—is essential for the initial untwisting of DNA and the subsequent ejection of the lagging strand from CMG. NS-EM analysis of the same Mcm2 6A mutant revealed that double hexamers can efficiently be converted to CMG (Fig. 4e–f); however, most fail to homo-dimerize and form complete dCMGE complexes (Fig. 4g). Hence, DNA binding by CMG is important for the stability of the dCMGE structure, in agreement with our observation that partial DNA digestion disrupts CMGE dimerization (Fig. 2e). The discovery that a MCM mutant that is unable to untwist DNA and support origin-dependent replication is competent in single

but not double CMG formation supports a functional role for CMG dimerization during replication initiation.

## Open DNA stabilized as the MCM dimer splits

CMG assembly leads to the disruption of the double hexamer interface, but how this is linked to the ATPase state of MCM is unclear[1]. When comparing the double hexamer and the dCMGE structures, we observed that the Mcm6 wedge insertion, which stabilizes the lagging-strand orphan bases in the dCMGE complex, is retracted and contributes to stabilizing the dimerization interface in the double hexamer (Fig. 5a). Residues T423 and R424 in the wedge insertion are surface-exposed and face on the outer perimeter of MCM in the double hexamer. As the DNA becomes untwisted, the Mcm6 wedge insertion disengages from the double hexamer interface and enters the helicase ring lumen in the dCMGE complex (Fig. 5b,c and Supplementary Video 5). In agreement with this observation, a combined T423E/R424E mutation (Mcm6 2E) supports MCM loading onto origin DNA to wild-type levels but negatively affects replication (Fig. 1d,f). Specific interactions between the ATPase and the N-terminal tiers in MCM reveal long-range allosteric changes that couple the ATPase state with the movement of the Mcm6 wedge. For example, the Mcm4 h2i ATPase element appears to act as a safety latch that keeps the Mcm6 wedge retracted in the double

hexamer. With the double-hexamer-to-dCMGE transition, a rigid-body rotation in the ATPase domains of Mcm4 and Mcm7 releases this latch, which creates enough space for the Mcm6 wedge insertion to disengage from the double hexamer interface and invade the helicase central channel (Fig. 5d and Supplementary Video 5). By compromising the Mcm4–6 latch interaction through the addition of five glutamate point mutations in the Mcm6 pore loop (T408E, Q409E, L410E, G411E and L412E), we generated an MCM variant (Mcm6 5E) that can be loaded to wild-type levels but is significantly defective for replication, possibly because it uncouples the ATPase state from the reconfiguration of the Mcm6 wedge (Fig. 1d,f). By combining our comparative structural and mutagenesis analyses, we propose a model whereby changes in the ATPase state promote dCMGE complex formation, which in turn couples melting of duplex DNA and splitting of the double hexamer. ADP release and ATP binding in the ATPase tier promotes the concerted movement of the h2i pore loops. Amongst these, Mcm2 h2i pulls on the leading-strand and pushes on the lagging-strand template, promoting duplex DNA untwisting, whereas Mcm4 h2i releases a latch that pins the N-terminal Mcm6 wedge insertion that is packed at the double hexamer homo-dimerization interface. As the latch is released, the Mcm6 wedge can swing inside of the MCM pore and stabilize the orphan bases that become exposed after the disruption of DNA base pairing in the untwisted DNA duplex. Supplementary Video 5 describes these structural transitions.

## Discussion

To ensure that replication occurs bidirectionally, several steps along the origin activation pathway in eukaryotic cells occur symmetrically. Symmetry is first established in G1 with the concerted and sequential loading of an inactive double hexamer[7,8]. During this process, origin loading of a first MCM hexamer creates a binding site for the symmetric loading of the second hexamer[24]. After entry into S phase, recruitment of firing factors that activate MCM depends on the phosphorylation of MCM by DDK (refs. [11,35–38]), which selectively targets fully loaded helicases by recognizing the symmetric structure of the double hexamer[39]. After the establishment of the replication fork, one of the two strands of the origin duplex is ejected from each CMG, and becomes the translocation strand of the opposing CMG. This symmetric fail-safe mechanism ensures that replication starts only if both helicases have been fully activated[1,40]. By imaging CMG caught in the act of nucleating origin DNA melting, we have identified yet another symmetric event on the path to origin activation. In fact, although CMGE formation disrupts the double hexamer interface, we found that the complex maintains a two-fold symmetric character, by forming a CMG dimer that is stabilized by both protein–protein interactions and DNA gripping. The CMGE dimer provides a head-to-head roadblock that limits ATPase-powered unidirectional translocation before the lagging strand is ejected, thus explaining previous observations that CMG formation and DNA untwisting at origins require ATP binding but not hydrolysis[1]. A CMGE dimer also explains why the CMG assembled around the origin DNA duplex during initiation is protected from disassembly before lagging-strand ejection[22]. In fact, whenever CMG transitions from engaging a fork to a DNA duplex, an MCM-binding site becomes accessible for the E3 ubiquitin ligase, SCF[Dia2] (in yeast, or CUL2[LRR1] in Metazoa), which sends the replisome to Cdc48-mediated disassembly. MCM ubiquitylation in duplex-engaged CMG occurs either upon termination of DNA replication or when the replisome engaged in fork progression encounters a nick on the lagging strand[17,19,20,41,42]. However, SCF[Dia2] cannot target the duplex-engaged replisome at initiation[22]. We now understand that this is because the dCMGE sterically impedes the docking of the E3 ligase onto MCM. In fact, when CMGE–SCF[Dia2] is superposed to our dCMGE structure, an extensive steric clash can be identified between the E3 ligase engaged to one ring and the Mcm3 subunit from the opposed ring in the CMG dimer (Extended Data Fig. 7).

The dCMGE nucleates two DNA bubbles inside each MCM ring, separated by 1.5 turns of duplex DNA, which might serve for the concerted recruitment of fork establishment factors, including Mcm10, RPA and Pol α. Mcm10 is known to trigger the ejection of the lagging strand and the ATPase-powered unwinding of the replication fork[1]. Although the mechanism of origin activation remains unknown, we note that Mcm10 engages the same N-terminal MCM elements[43,44] that mediate CMGE dimerization in our structural intermediate. A model for origin activation is presented in Extended Data Fig. 8 and further discussed in the Supplementary Discussion. Studies will be needed to establish whether Mcm10 engagement further disrupts the CMGE dimer interface, thereby releasing the inhibitory interaction that impairs ATPase-powered DNA translocation and allowing helicase bypass.

DNA replication, transcription and recombination all require the untwisting and opening of the double helix. Recent studies have described these processes in the transcription pre-initiation complex that supports RNA synthesis[45,46] and in the recombinases that promote strand exchange[46,47]. By contrast, the mechanism for the nucleation of DNA melting at an origin of replication has remained—to our knowledge—unknown for decades. Our work fills this gap. We describe the structure of the CMG replicative helicase assembled sequentially onto the *ARS1* origin, by reconstituting a multistep cellular process that involves 32 polypeptides[1]. Base-pair disruption involves ATP-triggered changes in MCM that promote pulling of the leading-strand and pushing of the lagging-strand template DNA. Our findings provide a framework in which to study replication initiation.

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

# Methods

## Cloning, expression and purification
ORC, Cdc6, Mcm2–7–Cdt1, DDK, CDK, Sld2, Sld3–Sld7, Cdc45, Dpb11, Pol ε, Pol ε exo-, Pol α, TopoI, Mcm10 and yeast histone octamer were purified on the basis of previously established protocols[1,11,24,33,48–51].

**Cloning, expression and purification of Mcm2–7–Cdt1 mutants.** Designed DNA fragments (Supplementary Table 1) were subcloned from pMA vectors (Supplementary Table 2) to pRS shuttle vectors (Supplementary Table 2), which were used to generate yeast strains (Supplementary Table 3) used to overexpress Mcm2–7–Cdt1 mutants. The oMG25 DNA fragment was subcloned from pMG39 to pAM38 using MluI and XbaI restriction sites to obtain pMG69, which was integrated into the yJF21 yeast strain, thus generating the yAE164 strain that was used to overexpress the Mcm2 6A mutant (Mcm2 V580A/K582A/P584A/K587A/W589A/K633A). The oMG27 DNA fragment was subcloned from pMG43 to pJF4 using BsiWI and SphI restriction sites to obtain pMG53, followed by the integration of pMG53 into the yAM20 strain, yielding the yAE160 strain, which was used for overexpression of the Mcm6 2E mutant (Mcm6 T423E/R424E). The oMG28 DNA fragment was subcloned from plasmid pMG44 to pJF4 using BsiWI and SphI restriction sites, thus obtaining plasmid pMG54. The pMG54 plasmid was integrated into the yAM20 strain, yielding the yAE161 strain that was used to overexpress the Mcm6 5E mutant (Mcm6 T408E/Q409E/L410E/G411E/L412E). All Mcm2–7–Cdt1 mutants were purified essentially as wild type[50].

**Cloning, expression and purification of GINS.** A gene block encoding a twin-strep tag and the first three codons of Psf3 was amplified and cloned into pFJD5 by restriction-free cloning techniques. A list of primers and gene blocks used is included in Supplementary Table 1. BL21(DE3)-CodonPlus-RIL cells (Agilent) were transformed with GINS expression plasmid (pJL003). Transformant colonies were inoculated into a 250-ml LB culture containing kanamycin (50 µg ml⁻¹) and chloramphenicol 35 µg ml⁻¹), which was grown overnight at 37 °C with shaking at 200 rpm. The following morning, the culture was diluted 100-fold into 6× 1 l of LB with kanamycin (100 µg ml⁻¹) and chloramphenicol (35 µg ml⁻¹). The cultures were left to grow at 37 °C until an optical density at 600 nm (OD$_{600 nm}$) of 0.5 was reached; 0.5 mM isopropyl β-D-1-thiogalactopyranoside (IPTG) was added to induce expression and cells were left shaking for 3 h. Cells were collected by centrifugation at 4,000 rpm for 20 min in a JS.4.2 rotor (Beckman). For lysis, cell pellets were resuspended in 120 ml of lysis buffer (100 mM Tris-HCl pH 8.0, 10% glycerol, 0.02% NP-40, 1 mM EDTA, 200 mM NaCl, Roche protease inhibitor tablets and 1 mM dithiothreitol (DTT) + 0.7 mM phenylmethylsulfonyl fluoride (PMSF). The lysate was sonicated for 120 s (5 s on, 5 s off) at 40% on a Sonics Vibra-Cell sonicator. Insoluble material was removed by centrifugation at 20,000 rpm for 30 min in a JS.25.50 rotor (Beckman). The supernatant was loaded by gravity onto a 1-ml Strep-TactinXT column (IBA). The resin was washed extensively with wash buffer (100 mM Tris-HCl pH 8.0, 10% glycerol, 1 mM DTT and 1 mM EDTA). GINS was eluted by the addition of 6 ml of 1× buffer BXT (IBA) supplemented with 10% glycerol and 1 mM DTT. The GINS-containing fractions were pooled and dialysed overnight in gel filtration buffer (25 mM HEPES-KOH pH 7.6, 10% glycerol, 0.02% NP-40, 200 mM potassium acetate and 1 mM DTT). The sample was concentrated and loaded onto a HiLoad 16/600 Superdex 200 equilibrated in the same buffer. GINS-containing fractions were pooled, aliquoted and snap-frozen in liquid N$_2$. About 22 mg GINS was purified from a 6-litre culture.

**Cloning, expression and purification of MH.** The codon-optimized expression sequence for MH containing a HRV 3C protease cleavage site followed by a twin-strep tag was synthesized and cloned into pET302 by GeneWiz Synthesis (pJL004). T7 express cells (NEB) were transformed with pJL004. Transformant colonies were inoculated into a 250-ml LB culture with ampicillin (100 µg ml⁻¹), which was grown overnight at 37 °C with shaking at 200 rpm. The following morning, the culture was diluted 100-fold into 6× 1 l of LB with ampicillin (100 µg ml⁻¹). The cultures were left to grow at 37 °C until an OD$_{600 nm}$ of 0.5 was reached; 0.5 mM IPTG was added to induce expression and cells were left shaking for 3 h. Cells were collected by centrifugation at 4,000 rpm for 20 min in a JS.4.2 rotor (Beckman). For lysis, cell pellets were resuspended in 80 ml of lysis buffer (20 mM Tris-HCl pH 8.5, 10% glycerol 0.5 mM EDTA, 500 mM KCl, Roche protease inhibitor tablets and 2 mM tris(2-carboxyethyl) phosphine (TCEP)) + 0.7 mM PMSF. The lysate was sonicated for 120 s (5 s on, 5 s off) at 40% on a Sonics Vibra-Cell sonicator. Insoluble material was removed by centrifugation at 20,000 rpm for 30 min in a JS.25.50 rotor (Beckman). The supernatant was loaded by gravity onto a 5-ml Strep-TactinXT column (IBA). The resin was washed extensively with lysis buffer. MH was eluted by the addition of 12 ml of 1× BXT (IBA) supplemented with 10% glycerol and 1 mM DTT. The MH-containing fractions were pooled and loaded onto a HiLoad 16/600 Superdex 75 equilibrated in gel filtration buffer (20 mM Tris-HCl pH 8.5, 10% glycerol 0.5 mM EDTA, 100 mM KCl and 0.5 mM TCEP). MH-containing fractions were pooled, aliquoted and snap-frozen in liquid N$_2$. About 36 mg MH was purified from a 6-litre culture.

## DNA templates
The native *ARS1* origin of replication flanked by Widom 601 and 603 sites or MH-flanked was amplified by PCR and purified as previously described[24]. The 6× *ARS1* array (pSSH005) was assembled by inserting an array of 6 *ARS1* origins with 40-bp spacing flanked by MH sites using NEBuilder HiFi assembly. The 6× *ARS1* origin array was amplified from pSSH005 using primer oSSH038 and concentrated by ethanol precipitation. A list of primers and DNAs used is included in Supplementary Table 1.

**Preparation and purification of chromatinized origin DNA.** Soluble yeast nucleosomes were reconstituted from octamers and DNA by salt gradient dialysis in several steps from 2 to 0.2 M NaCl as previously described[24]. Following nucleosome refolding, a final dialysis step was performed into loading buffer (25 mM HEPES-KOH pH 7.6, 80 mM KCl, 100 mM sodium acetate, 0.5 mM TCEP) and loaded onto a Superose 6 Increase 3.2/300 column equilibrated in the same buffer. Fractions containing *ARS1* origin DNA bound by 2 nucleosomes were pooled, concentrated, and stored at 4 °C. Reconstitution conditions were optimized by small-scale titration and nucleosomes checked by 6% native PAGE.

**Preparation and purification of MH-capped origin DNA.** *Short 168-bp MH-flanked origins.* The conjugation of MH with origin substrates was performed in 50 mM Tris-HCl pH 8.0, 1 mM EDTA and 0.5 mM 2-mercaptoethanol supplemented with 100 µM S-adenosylmethionine (NEB). The reaction was carried out overnight at 30 °C, with a 10:1 molar ratio of MH:DNA. After conjugation, reactions were centrifuged at 14,680 rpm for 5 min and loaded onto a 1 ml RESOURCE-Q column equilibrated into DNA buffer (50 mM Tris-HCl pH 8.0 and 5 mM 2-mercaptoethanol). MH-conjugated DNA was eluted in a linear gradient of DNA buffer B (50 mM Tris-HCl pH 8.0, 5 mM 2-mercaptoethanol and 2 M NaCl) over 24 column volumes. Fractions containing MH-conjugated DNA were pooled, concentrated and stored at −80 °C. Conjugations were checked by 6% native PAGE.

*6× ARS1 MH-flanked array.* The conjugation of MH with origin substrates was performed in 25 mM Tris-HCl pH 7.5, 10 mM magnesium acetate, 50 mM potassium acetate and 1 mg ml⁻¹ BSA supplemented with 150 µM S-adenosylmethionine (NEB). The reaction was carried out at 32 °C for 1 h then overnight at 4 °C, with a 20:1 molar ratio of MH:DNA. After

conjugation, reactions were centrifuged at 14,680 rpm for 5 min and loaded onto a Superose 6 Increase 10/300 column equilibrated into array buffer (25 mM HEPES-KOH pH 7.5, 200 mM NaCl and 1 mM DTT). Fractions containing MH-conjugated array DNA were pooled, concentrated and stored at 4 °C. Conjugations were checked by 6% native PAGE.

**616-bp *ARS1* circles.** The 616-bp *ARS1* circles were assembled and prepared as previously described[1] with the following modifications. The dephosphorylation step was performed with the use of quickCIP, instead of Antarctic phosphatase, for 30 min at 37 °C followed by enzyme inactivation at 80 °C for 2 min. After the ligation step, the DNA was concentrated as described and incubated with T5 exonuclease (NEB; 37 °C for 1 h) to eliminate non-ligated DNA. Ethanol precipitation, agarose electrophoresis and electroelution were omitted; instead, phenol/chloroform/isoamyl-alcohol extraction was performed, followed by ethanol precipitation using sodium acetate (pH 5.1) and the neutral carrier GeneElute Linear Polymer (LPA, MERCK).

## In vitro CMG assembly on short chromatinized origins

*ARS1* nucleosome-flanked origin DNA (20 nM) was incubated with 52 nM ORC, 52 nM Cdc6 and 110 nM Mcm2–7–Cdt1 for 30 min at 24 °C in loading buffer (25 mM HEPES-KOH pH 7.6, 100 mM potassium glutamate, 10 mM magnesium acetate, 0.02% NP-40 and 0.5 mM TCEP) + 5 mM ATP. The reaction was supplemented with 80 nM DDK, and incubation continued for a further 10 min at 24 °C. Nucleoprotein complexes were isolated by incubation with 5 µl MagStrep 'type3' XT beads (IBA) pre-washed in 1× loading buffer for 30 min at 24 °C. The beads were washed three times with 100 µl wash buffer (25 mM HEPES-KOH pH 7.6, 105 mM potassium glutamate, 5 mM magnesium acetate, 0.02% NP-40 and 500 mM NaCl) and once with 100 µl loading buffer. Loaded, phosphorylated double hexamers were eluted in 20 µl elution buffer (25 mM HEPES-KOH pH 7.6, 105 mM potassium glutamate, 10 mM magnesium acetate, 0.02% NP-40, 0.5 mM TCEP, 27 mM biotin and 5 mM ATP) for 10 min at 24 °C. The remaining supernatant was removed and incubated with 200 nM CDK for 5 min at 30 °C. A mix of firing factors was then added to a final concentration of 30 nM Dpb11, 100 nM GINS, 80 nM Cdc45, 20 nM Pol ε, 30 nM Sld3–Sld7 and 50 nM Sld2. After 30 min of incubation, the reaction was applied directly to grids or diluted fivefold in 1× loading buffer for ReconSil experiments.

## In vitro CMG assembly on 6× *ARS1* MH-capped array

MH-capped *ARS1* array DNA (5 nM) was incubated with 52 nM ORC, 52 nM Cdc6 and 110 nM Mcm2–7–Cdt1 for 30 min at 24 °C in loading buffer (25 mM HEPES-KOH pH 7.6, 100 mM potassium glutamate, 10 mM magnesium acetate, 0.02% NP-40 and 0.5 mM TCEP) + 5 mM ATP. The reaction was supplemented with 80 nM DDK, and incubation continued for a further 10 min at 24 °C. Nucleoprotein complexes were isolated by incubation with 5 µl MagStrep 'type3' XT beads (IBA) pre-washed in 1× loading buffer for 30 min at 24 °C. The beads were washed three times with 100 µl wash buffer (25 mM HEPES-KOH pH 7.6, 105 mM potassium glutamate, 5 mM magnesium acetate, 0.02% NP-40 and 500 mM NaCl) and once with 100 µl loading buffer. Loaded, phosphorylated double hexamers were eluted in 20 µl elution buffer (25 mM HEPES-KOH pH 7.6, 105 mM potassium glutamate, 10 mM magnesium acetate, 0.02% NP-40, 0.5 mM TCEP, 27 mM biotin and 5 mM ATP) for 10 min at 24 °C. The remaining supernatant was removed and incubated with 200 nM CDK for 5 min at 30 °C. A mix of firing factors was then added to a final concentration of 90 nM Dpb11, 300 nM GINS, 240 nM Cdc45, 60 nM Pol ε, 90 nM Sld3–Sld7 and 150 nM Sld2. After 30 min of incubation, the reaction was diluted fivefold in 1× loading buffer and applied to grids.

For experiments in which DNA was partially digested after the CMG formation reaction, MseI (NEB) was added at a concentration of 0.1 U diluted in 1× loading buffer. Incubation was performed for 10 min at 30 °C before applying to EM grids.

## In vitro DNA replication assays

Replication assays were performed as described previously[52]. The reactions were incubated in a ThermoMixer at 30 °C with 1,250 rpm shaking. The reaction buffer was as follows: 25 mM HEPES-KOH pH 7.6, 10 mM magnesium acetate, 2 mM DTT, 0.02% NP-40, 100 mM potassium glutamate and 5 mM ATP. MCM helicase loading reaction (5 µl) contained 30 nM ORC, 30 nM Cdc6, 60 nM Mcm2–7–Cdt1 (or MCM mutants) and either 4 nM ARS-containing 10.6 kb supercoiled plasmid (pJY22; Supplementary Table 2) or 40 nM ARS-containing short linear DNA (flanked by nucleosomes or MH; Supplementary Table 2) as for Fig. 1. After 20 min, DDK was added to a final concentration of 50 nM and further incubated for 20 min. Next, the reaction volume was doubled (final volume was 10 µl) by adding proteins (20 nM Pol ε, 30 nM Dpb11, 20 nM GINS, 50 nM Cdc45, 20 nM CDK, 50 nM RPA, 10 nM TopoI, 100 nM Pol α, 25 nM Sld3–Sld7, 10 nM Mcm10 and 50 nM Sld2) and nucleotides (200 µM CTP, 200 µM GTP, 200 µM UTP, 80 µM dCTP, 80 µM dGTP, 80 µM dTTP, 80 µM dATP and 50 nM $\alpha^{32P}$-dCTP). For replication reactions with linear DNA (Fig. 1) Pol ε exo- was used instead of Pol ε wild type to reduce end labelling and the concentration of deoxynucleotides was modified (that is, 30 µM dCTP, 30 µM dGTP, 30 µM dTTP, 30 µM dATP and 100 nM $\alpha^{32P}$-dCTP). The reactions were stopped by EDTA after 15 and 30 min for reactions with 10.6-kb supercoiled DNA or after 20 min for reactions with short linear DNA substrates and processed as described[51,52]. The replication products were separated using 0.8% agarose alkaline gel for 17 h at 25 V for reactions with 10.6-kb supercoiled DNA. For reactions with short DNA substrates, samples were separated using 2% agarose alkaline gel for 4 h at 38 V. The image signal from Fig. 1e was background-subtracted in Fiji using the subtract background algorithm in Fiji v.2.0.0 (ref. [53]).

## DNA topology assay

The experiment was performed as described previously[1]. The concentrations of proteins were as follows: 10 nM ORC, 50 nM Cdc6, 100 nM Mcm2–7–Cdt1 (or Mcm mutants), 80 nM DDK for the helicase loading step (5 µl) and 20 nM Pol ε, 30 nM Dpb11, 40 nM GINS, 50 nM Cdc45, 30 nM CDK, 10 nM TopoI, 25 nM Sld3–7, 5 nM Mcm10, 50 nM Sld2 for the helicase activation step (10 µl). Radiolabelled 616-bp circular DNA (25 fmol) was used. After processing the reactions as described previously[1], Ficoll 400 (final concentration was 2.5%) and Orange G were used to load the sample onto a native 3.5% bis-polyacrylamide gel (1× TBE) and separation was carried out for 21 h at 90 V using Protean II XL Cell apparatus (Bio-Rad) at room temperature. The 0.7-mm gel was dried (without fixation) at 80 °C for 105 min, exposed to a phosphor screen and scanned with the use of Typhoon phosphor imager.

## Sample preparation and data collection for NS-EM

NS-EM sample preparation was performed on 400-mesh copper grids with carbon film (Agar Scientific). Grids were glow-discharged for 30 s at 45 mA using a K100X glow discharge unit (Electron Microscopy Sciences) before a 4-µl sample was applied to the grids and incubated for 2 min. Grids were stained by two successive applications of 4 µl 2% (w/v) uranyl acetate with blotting between the first and second application. Stained grids were blotted after 20 s to remove excess stain. Unless described otherwise, data collection was carried out on a Tecnai LaB6 G2 Spirit transmission electron microscope (FEI) operating at 120 keV. A 2K × 2K GATAN Ultrascan 100 camera was used to collect micrographs at a nominal magnification of 30,000 (with a physical pixel size of 3.45 Å per pixel) within a −0.5 to −2.0 µm defocus range.

## NS-EM image processing

A subset of particles was manually picked using RELION-3.1 (ref. [26]) and used as a training dataset for Topaz training[53]. Subsequent image processing was performed using RELION-3.1. The CTF of each micrograph was estimated using Gctf (ref. [54]) and particles

were extracted and subjected to reference-free 2D classification in RELION-3.1.

## ReconSil image processing
For ReconSil experiments, image processing was carried out as detailed above. Reference-free 2D classification in RELION generates both 2D class averages and star files detailing the class assignment, particle coordinates and transformations (translations and rotations) applied to the raw particles for alignment. 2D averages are superposed on the raw micrographs, overlaid on the particles that contributed to their generation. This yielded signal-enhanced 'ReconSiled' micrographs reconstituting the context of complete origins of replication. ReconSiled micrographs were used for the selection and rejection of origin nucleoproteins for further analysis.

## ReconSil data analysis and statistics
ReconSiled origins were analysed as previously described[24]. In brief, ReconSiled micrographs were used to re-extract particles of interest in RELION. Selected particles were manually classified for statistical analysis. Measurements of ReconSiled origins were performed manually using Fiji[55] and plotted in GraphPad Prism v.9.2.0.

## Sample preparation and data collection for cryo-EM
CMG assembly reactions (reconstituted as described in 'In vitro CMG assembly on short chromatinized origins') were frozen on 400-mesh lacey grids with a layer of ultra-thin carbon (Agar Scientific). All grids were freshly glow-discharged for 1 min at 45 mA using a K100X glow discharge unit (Electron Microscopy Sciences) before plunge freezing. Samples were prepared by applying 4 μl of undiluted CMG assembly reactions for 2 min on a grid equilibrated to 25 °C in 90% humidity. The grid was blotted for 4.5 s and plunged into liquid ethane. Data collection was performed on an in-house Thermo Fisher Scientific Titan Krios transmission electron microscope operated at 300 kV, equipped with a Gatan K2 direct electron detector camera (Gatan) and a GIF Quantum energy filter (Gatan). Images were collected automatically using the EPU software (Thermo Fisher Scientific) in counting mode with a physical pixel size of 1.08 Å per pixel, with a total electron dose of 51.4 electrons per Å$^2$ during a total exposure time of 10 s dose-fractionated into 32 movie frames (Extended Data Table 1). We used a slit width of 20 eV on the energy filter and a defocus range of −2.0 to −4.4 μm. A total of 65,286 micrographs were collected from two separate sessions.

## Cryo-EM image processing
Data processing was performed using RELION-3.1 (ref. [26]) and cryoSPARC v.3.2 (ref. [56]) (Extended Data Fig. 3). The movies for each micrograph were first corrected for drift and dose-weighted using MotionCorr2 (ref. [57]). CTF parameters were estimated for the drift-corrected micrographs using Gctf within RELION-3.1 (ref. [54]). Dataset one was first processed separately and combined with dataset two at a later stage.

For the first dataset, particles were picked using a manually curated particle set as a template in crYOLO v.1.7.5 (ref. [58]). These particles were binned by 2 and extracted with a box size of 360 pixels for 2D and 3D classification. A subset of 1,600 representative particles across the entire defocus range was selected. Picks in areas of obvious particle aggregation were removed along with particles located on the carbon lace. A Topaz[53] model was then iteratively trained on the remaining particles. All particles were re-picked with the Topaz model with the default score threshold of 0 for particle prediction. The two datasets were combined and a total of 927,109 particles were picked, binned by 2 and extracted with a box size of 360 pixels. We carried out 2D classification to remove remaining smaller particles and contaminants. We subjected the remaining particles to 3D multi-reference classification with four sub-classes, angular sampling of 7.5°, a regularization parameter $T$ of 5 using low-pass-filtered initial models from previous ab initio and processing steps on dataset 1 of dCMGE complexes, and double hexamer model generated from EMD-3960 (Extended Data Fig. 3). The resulting 133,262 (trans-dCMGE) and 46,049 (cis-dCMGE) particles with density corresponding to Pol ε on both CMG molecules were un-binned and refined to yield maps with resolutions of 7.7 and 14.4 Å. C2 symmetry imposition did not improve the quality of the maps. The 133,262 trans-dCMGE particles were imported into cryoSPARC and subjected to multiple rounds of non-uniform refinement, heterogenous 3D classification and non-uniform local refinement, yielding a map at approximately 8 Å (Extended Data Fig. 3). Attempts to improve cis-dCMGE were unsuccessful given the limited particle numbers. As expected, these reconstructions do not show secondary structural features owing to the conformational heterogeneity between the two CMGE molecules bound by flexible DNA. We applied a C2 symmetry expansion procedure to both trans- and cis-dCMGE particles (179,311) with re-centring on one CMGE in RELION and combined all particles. We also downsized the box size to 512 pixels during this process to speed up downstream processing. Following this, masked 3D refinement with local searches in C1 of the centred single CMGE (consisting of 358,622 particles) was refined to 4.2-Å resolution. These particles were subjected to several rounds of CTF refinement and two rounds of Bayesian polishing. After this, CTF-refined and polished particles were refined with local searches in C1 with a mask encompassing the entire CMGE density to 3.6-Å resolution. To better resolve the DNA inside the MCM central channel, densities corresponding to Cdc45, GINS and Pol ε were subtracted in RELION. Signal-subtracted particles were analysed by 3D variability analysis in cryoSPARC (ref. [56]). A subset of 71,348 particles was selected based on the quality of DNA density. These signal-subtracted particles were subsequently reverted to the original particles and refined using local searches in C1 using local searches to 3.5-Å resolution.

All refinements were performed using fully independent data half-sets and resolutions are reported based on the Fourier shell correlation (FSC) = 0.143 criterion (Extended Data Fig. 2). FSCs were calculated with a soft mask. Maps were corrected for the modulation transfer function of the detector and sharpened by applying a negative B-factor as determined by the post-processing function of RELION or in cryoSPARC. The final RELION half-maps were used to produce a density modified map using the PHENIX Resolve CryoEM (refs. [28,59]). This 3.4-Å map showed significant improvements for side chain and DNA density as well as for overall interpretability. Local-resolution estimates were determined using PHENIX or cryoSPARC (Extended Data Fig. 2f,j). The conversions between cryoSPARC and RELION files were performed using the UCSF pyem v.0.5 package[60].

## Model building and refinement
CMG (from PDB 6SKL)[31], Pol2 subunit (from PDB 6HV9)[33] and a homology model of the N-terminal domain of Dpb2 obtained from the Phyre2 server[61] were docked initially into the cryo-EM map produced from Resolve CryoEM, using USCF Chimera, and refined against the map using Namdinator[62] as a starting point for modelling with Coot v.0.9.1 (ref. [63]). The DNA and the MCM5 winged helix domain were built de novo. The register of origin DNA engagement of dCMGE is heterogeneous because MCM double hexamers can slide along duplex DNA before dCMGE is formed. For this reason we could not build the origin DNA sequence with certainty and modelled polyA:polyT DNA instead. The resulting model was then subjected to an iterative process of real-space refinement using Phenix.real_space_refinement[64] with geometry and secondary structure restraints and base-pairing and base-stacking restraints where appropriate, followed by manual inspection and adjustments in Coot. The geometries of the atomic model were evaluated by the MolProbity webserver[65].

## Map and model visualization
Maps were visualized in UCSF Chimera[66] and ChimeraX[67] and all model illustrations and morphs were prepared using ChimeraX or PyMOL.

## Statistics and reproducibility

Statistical analysis was performed using a two-tailed Welch's *t*-test in GraphPad Prism v.9.2.0. No statistical methods were used to pre-determine sample size. The experiments were not randomized, and investigators were not blinded to allocation during experiments and outcome assessment.

## Reporting summary

Further information on research design is available in the Nature Research Reporting Summary linked to this paper.

## Data availability

Data supporting the findings of this study are available within the paper and its Supplementary Information files. Cryo-EM density maps of the CMGE dimer complex have been deposited in the Electron Microscopy Data Bank (EMDB) under the accession code EMD-13988. The cryo-EM density map of the symmetry-expanded CMGE monomer has been deposited in the EMDB under the accession code EMD-13978. Atomic coordinates have been deposited in the PDB with the accession codes 7QHS (symmetry-expanded CMGE monomer) and 7Z13 (monomer docked into the CMGE dimer map).

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

**Acknowledgements** We thank past and present members of the A.C. and J.F.X.D. laboratories for discussions; T. Miller for sharing the strep-tagged yeast histone H3 construct; J. Locke for help screening cryo grids; M. Douglas, A. Cheung, O. Acton, P. Rosenthal and V. Pye for discussions; B. Canal for Pol ε exo-; and A. Early for helping M.H.G. with the transformation of Mcm6 and Mcm2 mutants. We acknowledge the Crick Structural Biology STP for support on the screening microscope (E. Punch); computational support (A. Purkiss and P. Walker); and yeast cultures (N. Patel, A. Alidoust and D. Patel). Thanks to A. Noble and T. Bepler for advice using Topaz. This work was funded jointly by the Wellcome Trust, MRC and CRUK at the Francis Crick Institute (FC001065 and FC001066). A.C. receives funding from the European Research Council (ERC) under the European Union's Horizon 2020 research and innovation programme (grant agreement no. 820102). J.S.L., J.S. and M.H.G are the recipients of a European Molecular Biology Organization (EMBO) long-term fellowship award (ALTF 211–2020, 1177–2020 and 34–2021). S.S.H. is a recipient of a Human Frontier Science Program (LT000834/2020-L) and EMBO long-term fellowship award (ALTF 962–2019). This work was also funded by a Wellcome Trust Senior Investigator Award (106252/Z/14/Z) and a European Research Council Advanced Grant (669424-CHROMOREP) to J.F.X.D.

**Author contributions** J.S.L. and A.C. conceived the study. J.S.L. and M.H.G. designed biochemistry experiments. J.S.L., S.S.H. and J.F.G. prepared biochemical reagents and developed all assays, with the exception of the DNA replication and topology experiments that were performed by M.H.G., who also produced all MCM mutants under the supervision of J.F.X.D. J.S.L. and S.S.H. performed NS-EM imaging. J.S.L., A.C. and A.N. performed cryo-EM imaging. J.S.L. conducted all image processing. J.S.L. performed the atomic model building and validation with J.S. J.F.X.D. contributed critical reagents and protocols. A.C. supervised the study. J.S.L. and A.C. wrote the manuscript with input from M.H.G., J.F.X.D. and the other authors.

**Competing interests** The authors declare no competing interests.

**Additional information**
**Correspondence and requests for materials** should be addressed to Alessandro Costa.

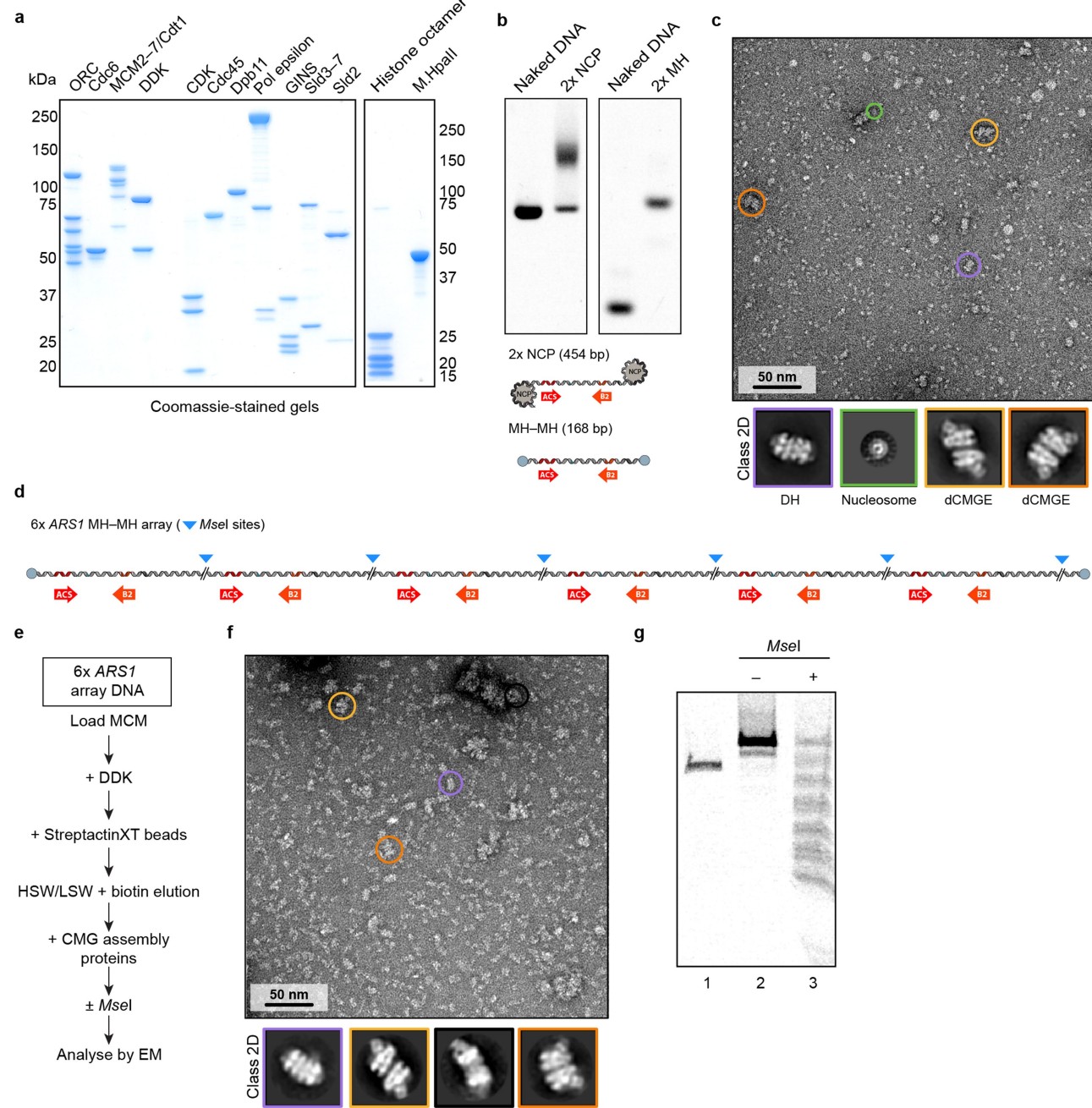

**Extended Data Fig. 1 | Origin-dependent CMG assembly with purified proteins visualized by electron microscopy. a**. Purified MCM loading and firing factors (left), and additional factors required for DNA substrate preparation (right) analysed by SDS–PAGE with Coomassie staining. For gel source data, see Supplementary Fig. 1. Similar results were observed for at least two independent sample preparations. **b**. 6% PAGE gel of capped, origin DNA substrates used in this study. (Below) Cartoon representation of *ARS1* origins of replication, containing the two inverted ORC-binding sites, ACS (high affinity, red arrow) and B2 (low affinity, orange arrow). *ARS1* is flanked by nucleosomes (NCP) or covalently linked methyltransferases (MH). For gel source data, see Supplementary Fig. 1. Similar results were observed for three independent sample preparations. **c**. Representative NS micrograph and 2D averages of entire CMG assembly reactions used to generate the ReconSiled origins shown in Fig. 1b. 70% of CMG particles exist in a dimeric (dCMGE) *trans* configuration (light orange), with GINS positioned on opposed sides of MCM. 11% of dCMGE

particles exist in a *cis* configuration (dark orange) that might derive from *trans*-dCMGE disengagement and rotation. This experiment was performed more than three times. **d**. Cartoon representation of 6x *ARS1* array, containing the two inverted ORC-binding sites, ACS (high affinity, red arrow) and B2 (low affinity, orange arrow). Each *ARS1* origin is separated by 40 bp linker DNA. Array is flanked by covalently attached MH. Blue arrows indicate *Mse*I cut sites. **e**. Reaction scheme for CMG assembly reactions on DNA substrates containing a 6x *ARS1* array. **f**. Representative NS micrograph and representative double hexamer and dCMGE 2D averages obtained from CMG assembly reaction on 6× *ARS1* array. This experiment was performed three times. **g**. 6% PAGE gel of partial DNA digestion of 6x ARS array by *Mse*I carried out under the same conditions as NS-EM experiments. Lane 1 contains unmodified 6× *ARS1* array. Lane 2 contains MH-conjugated 6x *ARS1* array DNA. Lane 3 contains *Mse*I digested 6× *ARS1* array DNA. For gel source data, see Supplementary Fig. 1. This experiment was performed twice.

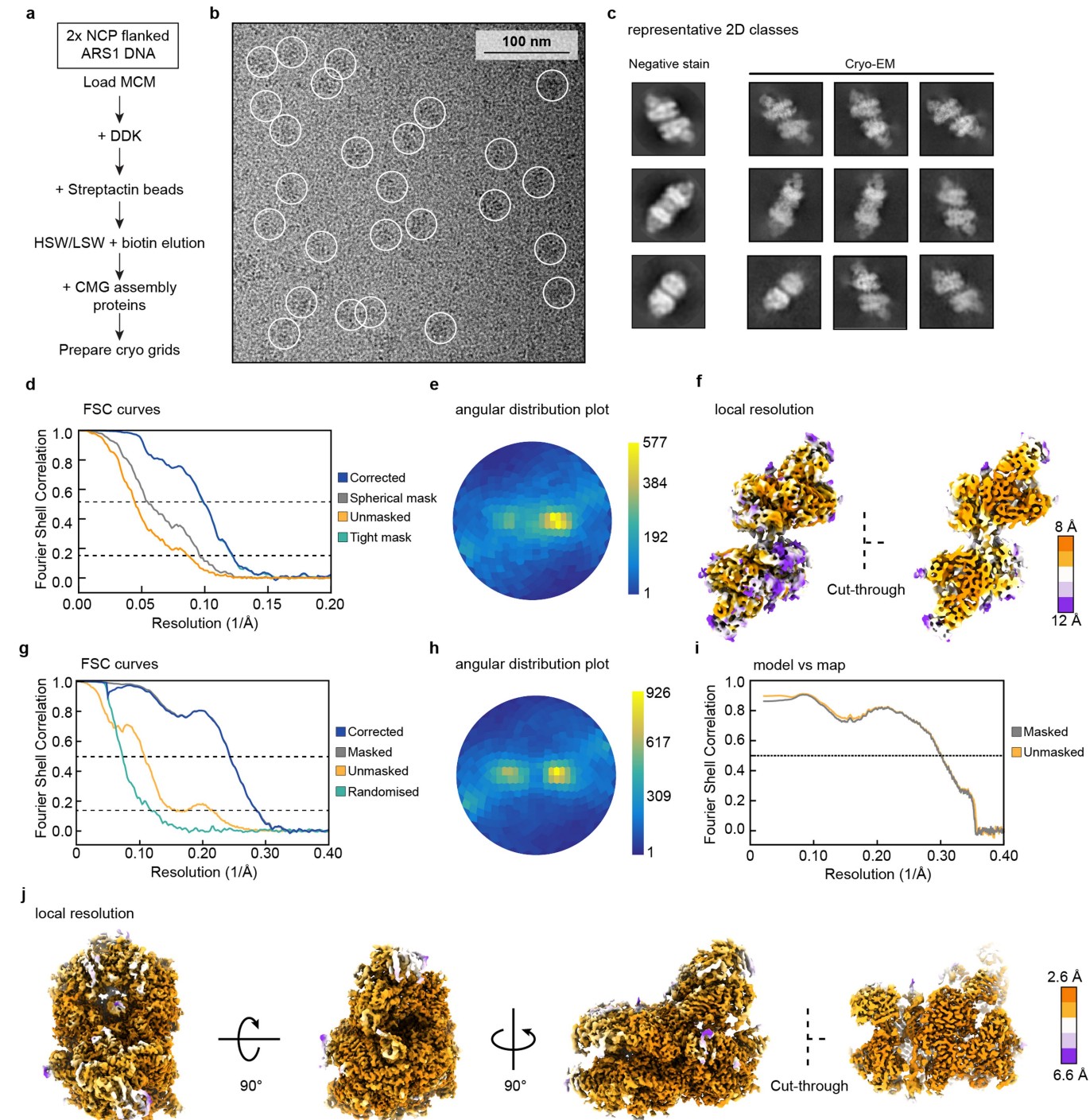

**Extended Data Fig. 2 | Sample preparation and validation of dCMGE cryo-EM reconstructions. a.** Schematic of biochemical reconstitution used for cryo-EM samples. **b.** Representative cryo-EM micrograph of entire dCMGE assembly reaction with particles highlighted with white circles. Cryo-EM sample preparation was performed once; similar results were observed in at least three independent NS sample preparations. b **c.** Representative 2D class averages from NS-EM (left panel) and cryo-EM imaging (right panel).

Box widths represent 500 Å. **d.** Fourier shell correlation plot for the C1-refined CMGE dimer map. **e.** Angular distribution plot for the C1-refined CMGE dimer map. **f.** CryoSPARC local-resolution estimate for the C1-refined CMGE dimer map. **g.** Fourier shell correlation plot for symmetry-expanded CMGE map used in model building. **h.** Angular distribution plot for the symmetry-expanded CMGE map used in model building. **i.** Model-to-map correlation graph. **j.** PHENIX local-resolution estimate for the symmetry-expanded CMGE map.

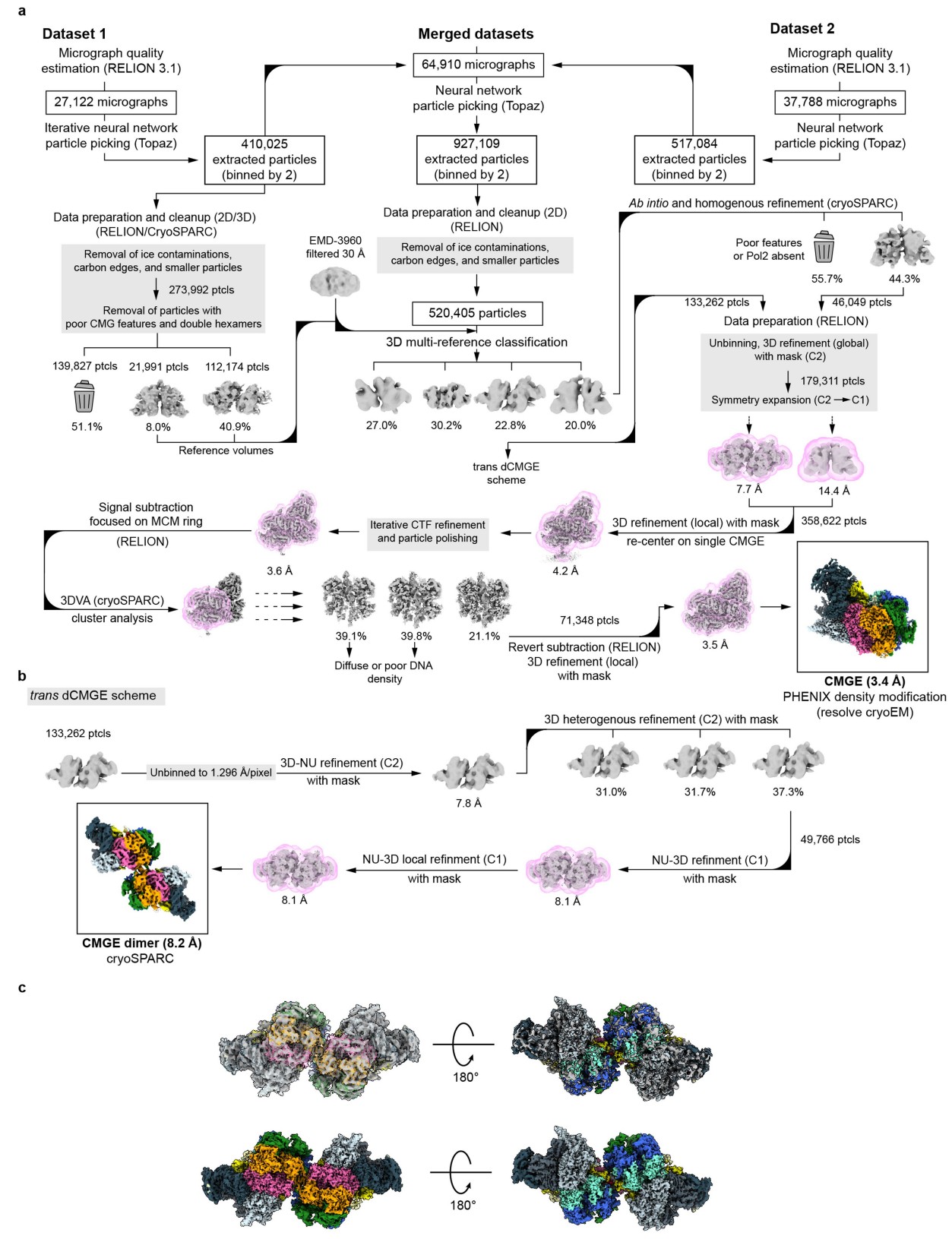

**a**

**Dataset 1**

Micrograph quality estimation (RELION 3.1)

27,122 micrographs

Iterative neural network particle picking (Topaz)

**Merged datasets**

64,910 micrographs

Neural network particle picking (Topaz)

**Dataset 2**

Micrograph quality estimation (RELION 3.1)

37,788 micrographs

Neural network particle picking (Topaz)

410,025 extracted particles (binned by 2)

927,109 extracted particles (binned by 2)

517,084 extracted particles (binned by 2)

Data preparation and cleanup (2D/3D) (RELION/CryoSPARC)

Removal of ice contaminations, carbon edges, and smaller particles

273,992 ptcls

Removal of particles with poor CMG features and double hexamers

139,827 ptcls    21,991 ptcls    112,174 ptcls

51.1%    8.0%    40.9%

Reference volumes

EMD-3960 filtered 30 Å

Data preparation and cleanup (2D) (RELION)

Removal of ice contaminations, carbon edges, and smaller particles

520,405 particles

3D multi-reference classification

27.0%    30.2%    22.8%    20.0%

trans dCMGE scheme

*Ab intio* and homogenous refinement (cryoSPARC)

Poor features or Pol2 absent

55.7%    44.3%

133,262 ptcls    46,049 ptcls

Data preparation (RELION)

Unbinning, 3D refinement (global) with mask (C2)

179,311 ptcls

Symmetry expansion (C2 → C1)

7.7 Å    14.4 Å

358,622 ptcls

Signal subtraction focused on MCM ring (RELION)

Iterative CTF refinement and particle polishing

3D refinement (local) with mask re-center on single CMGE

4.2 Å

3.6 Å

3DVA (cryoSPARC) cluster analysis

39.1%    39.8%    21.1%

Diffuse or poor DNA density

71,348 ptcls

Revert subtraction (RELION) 3D refinement (local) with mask

3.5 Å

**CMGE (3.4 Å)**
PHENIX density modification (resolve cryoEM)

**b**

*trans* dCMGE scheme

133,262 ptcls

Unbinned to 1.296 Å/pixel

3D-NU refinement (C2) with mask

7.8 Å

3D heterogenous refinement (C2) with mask

31.0%    31.7%    37.3%

49,766 ptcls

NU-3D local refinement (C1) with mask

8.1 Å

NU-3D refinment (C1) with mask

8.1 Å

**CMGE dimer (8.2 Å)**
cryoSPARC

**c**

180°

180°

**Extended Data Fig. 3 | Cryo-EM data processing pipeline for dCMGE assembled on nucleosome-capped origin DNA. a**. Schematic shows the classification and refinement steps taken to achieve the symmetry-expanded refined CMGE cryo-EM structure. The software used during each processing step is listed. **b**. Schematic of the classification and refinement pipeline for the C1-refined CMGE dimer. **c**. Symmetry-expanded refined CMGE structure (coloured) docked into C1-refined CMGE dimer map (grey). Reported resolutions in all schematics are calculated based on the FSC = 0.143 criterion.

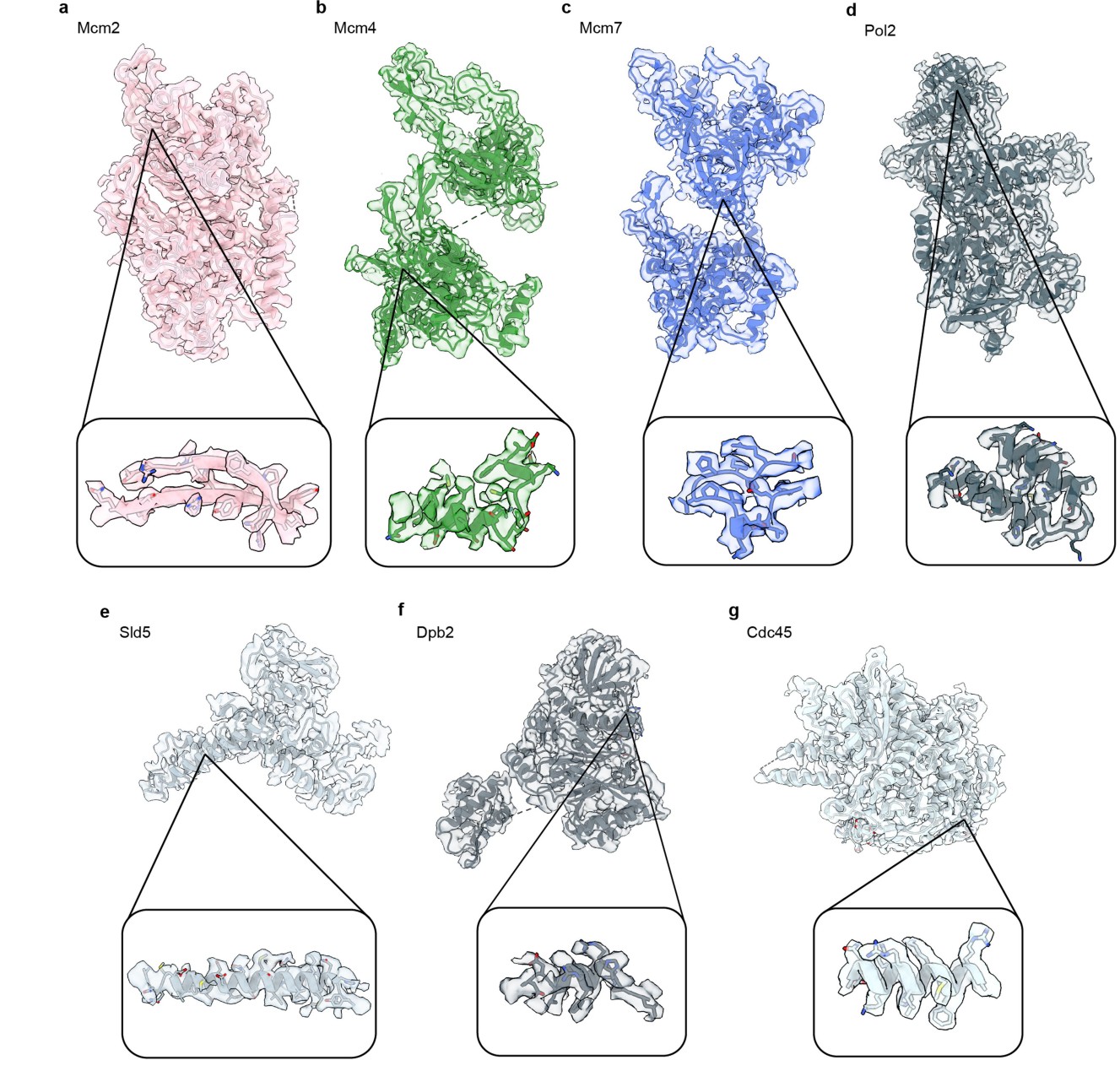

**Extended Data Fig. 4 | Quality of cryo-EM densities.** Example cryo-EM density of Mcm2 (**a**), Mcm4 (**b**), Mcm7 (**c**), Pol2 (**d**), Sld5 (**e**), Dpb2 (**f**), and Cdc45 (**g**). For each subunit, side chain density features are shown in the inset.

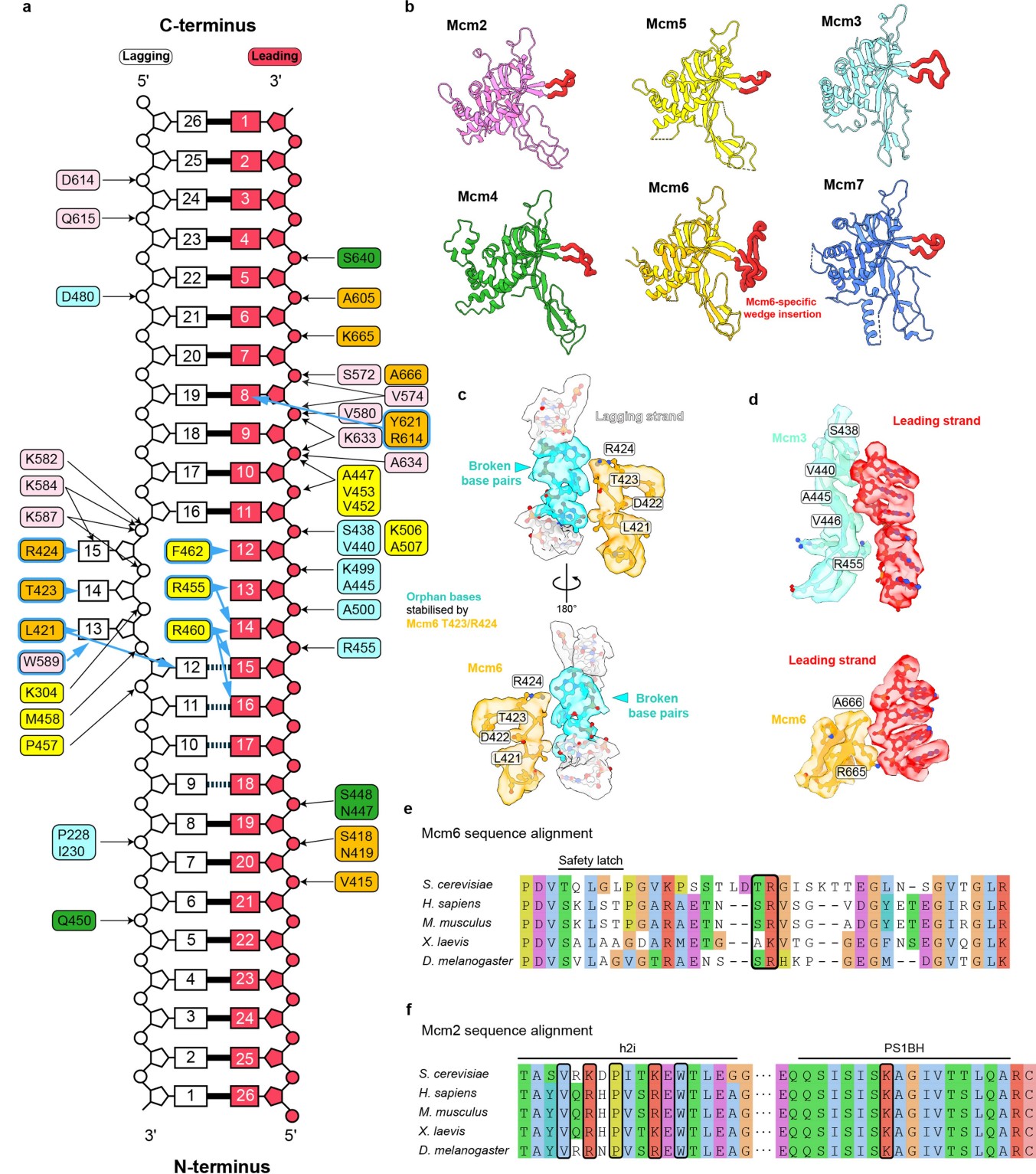

**Extended Data Fig. 5 | Analysis of protein–DNA interactions within the dCMGE complex. a**. MCM contacts with DNA in dCMGE. Phosphate backbone interactions are indicated with black arrows and base interactions are highlighted with blue arrows. Residues are coloured based on individual MCM subunits. **b**. Comparison of N-terminal domains of Mcm2–7. The N-terminal pore loops are highlighted in red. Mcm6 contains a unique insertion ('wedge') with residues that stabilize the orphan bases exposed upon DNA untwisting.

**c**. Cryo-EM density of flexible Mcm6 wedge stabilizing three lagging-strand bases. **d**. Representative cryo-EM densities of DNA contacts for Mcm3 and Mcm6. **e**. Sequence alignment of N-terminal region of Mcm6. Conserved T423 and R424 are outlined with black boxes. **f**. Sequence alignment between the h2i and PS1BH motifs of yeast Mcm2 compared to selected higher eukaryotes. Mcm2 6A mutations are outlined with black boxes. All alignments are coloured using the ClustalX colouring scheme.

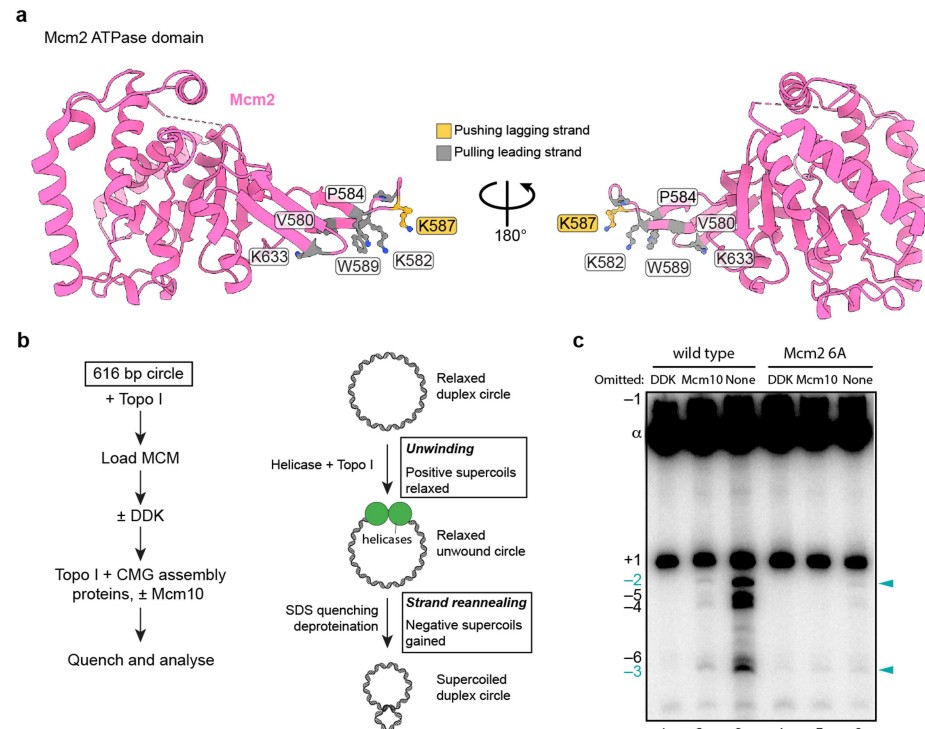

**Extended Data Fig. 6 | Supporting information for the mechanism of DNA-bubble nucleation. a**. Overview of the Mcm2 6A mutant in the context of ATPase domain. Residues the push the lagging-strand template are coloured gold and residues that pull the leading-strand template are coloured in grey. **b**. Reaction scheme for topology footprint assay for DNA unwinding. **c**. Over-exposed gel as seen in Fig. 4d. For gel source data, see Supplementary Fig. 1. This experiment was performed twice.

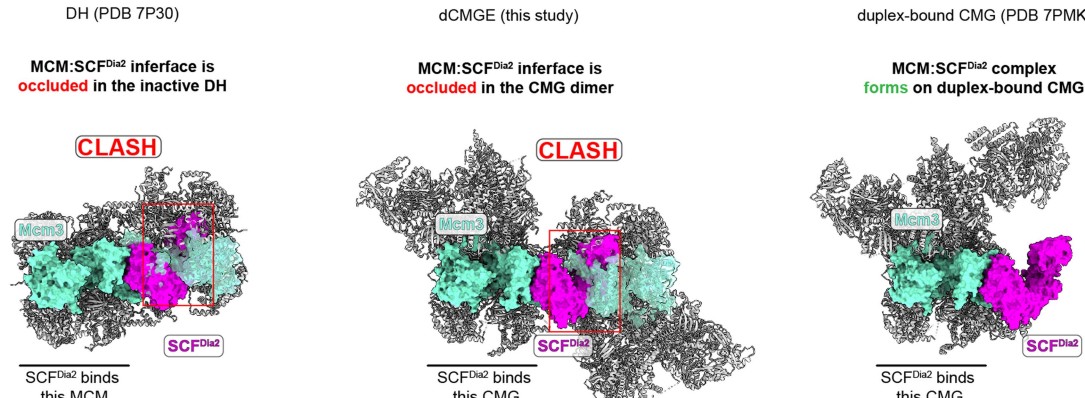

**Extended Data Fig. 7 | dCMGE sterically impedes the docking of the E3 ligase onto MCM.** When CMGE–SCF$^{Dia2}$ (7PMK) is superposed onto the double hexamer (7P30) and the dCMGE structure (this study), major clashes can be identified between SCF$^{Dia2}$ engaged to one ring and the Mcm3 subunit from the opposed ring in the CMG dimer. This clash explains why the CMG assembled around the origin DNA duplex during initiation is protected from disassembly before lagging-strand ejection[22].

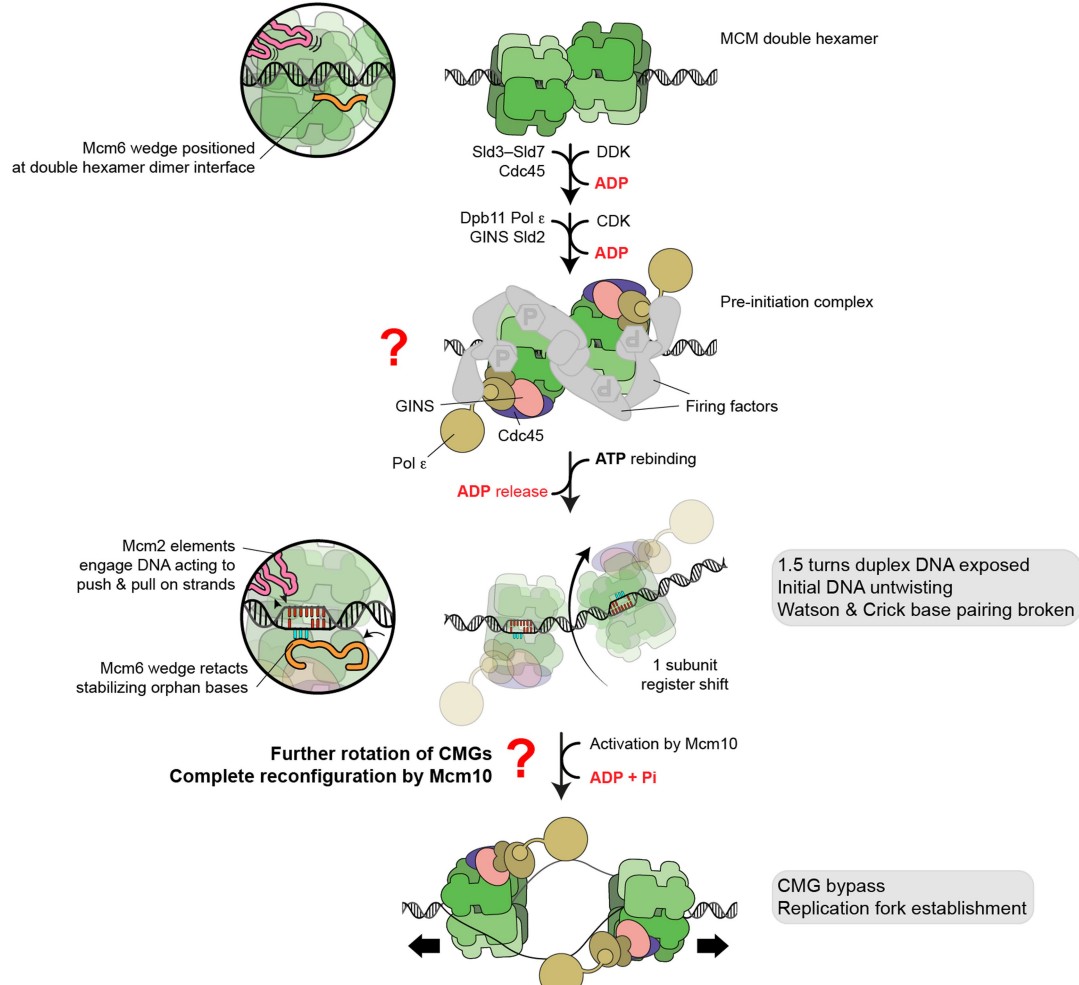

**Extended Data Fig. 8 | Schematic representation of the steps that lead to replication origin firing.** From top to bottom: double hexamer is loaded onto duplex origin DNA in an ATP-hydrolysis dependent manner. ADP formed during double hexamer assembly remains bound to MCM. The Mcm6 wedge insertion forms part of the double hexamer dimerization interface (inset). After loading the double hexamer makes limited contacts with duplex DNA. DDK phosphorylation triggers the recruitment of firing factors that deposit Cdc45 (C in CMGE) and GINS (G) onto the MCM (M), in the context of the so-called pre-initiation complex, the formation of which requires Pol ε (E) and CDK kinase activity. ADP release and binding of new ATP by MCM triggers CMGE assembly. CMGE assembly leads to the reconfiguration of the double hexamer interface, resulting in hexamer separation and a 1 subunit register shift pivoting around the Mcm6 N-terminal domain. This movement results in the exposure of 1.5 turns of duplex DNA between two MCM rings, and nucleation of DNA melting within each MCM ring. DNA melting is promoted upon ATP-triggered untwisting of 0.7 turns of the DNA, through the action of Mcm2 (pink in the inset), which pushes on the lagging strand while simultaneously pulling on the leading strand. As DNA is untwisted, Watson and Crick base pairs are broken, and three orphan bases become stabilized by the Mcm6 wedge (orange in the inset), which moves from the double hexamer interface and invades the MCM central channel. Action by Mcm10 triggers ATP hydrolysis by CMG and ejection of the lagging strand through an unknown mechanism, resulting in helicase bypass and establishment of replication forks.

**Extended Data Table 1 | Cryo-EM data collection, refinement and validation statistics**

| | Symmetry expanded CMGE (EMDB-13978) (PDB 7QHS) | | CMGE dimer (EMD-13988) |
|---|---|---|---|
| **Data collection and processing** | | | |
| Magnification | 130,000x | | 130,000x |
| Voltage (kV) | 300 | | 300 |
| Electron exposure (e–/Å²) | 51.4 | | 51.4 |
| Defocus range (μm) | 2.0–4.4 | | 2.0–4.4 |
| Pixel size (Å) | 1.08 | | 1.08 (collection) 1.296 (processing) |
| Symmetry imposed | C1 | | C1 |
| Initial particle images (no.) | 927,109 | | 133,262 |
| Final particle images (no.) | 71,348 | | 49,766 |
| Map resolution (Å) | | | |
| RELION | 3.50 | CryoSPARC | 8.20 |
| FSC threshold | 0.143 | | 0.143 |
| Resolve Cryo-EM | 3.40 | | |
| FSC threshold | 0.5 | | 0.5 |
| Map resolution range (Å) | 2.6–6.6 | | 8.0–12.1 |
| | | | |
| **Refinement** | | | |
| Initial model used (PDB code) | 6SKL 6HV9 | | |
| Model resolution (Å) | | | |
| FSC threshold = 0.5 | 3.30 | | |
| Map sharpening $B$ factor (Å²) | Resolve Cryo-EM (–10) | | |
| Model composition | | | |
| Non-hydrogen atoms | 53,679 | | |
| Protein residues | 6,564 | | |
| Nucleotides | 52 | | |
| Ligands | ZN: 7 MG: 3 ATP: 4 ADP: 2 | | |
| $B$ factors (Å²) | | | |
| Protein | 38.81 | | |
| Nucleotide | 128.08 | | |
| Ligand | 40.25 | | |
| R.m.s. deviations | | | |
| Bond lengths (Å) | 0.003 | | |
| Bond angles (°) | 0.645 | | |
| Validation | | | |
| MolProbity score | 1.65 | | |
| Clashscore | 7.14 | | |
| Poor rotamers (%) | 0.03 | | |
| Ramachandran plot | | | |
| Favored (%) | 96.20 | | |
| Allowed (%) | 3.80 | | |
| Disallowed (%) | 0.00 | | |

# Reporting Summary

## Statistics

For all statistical analyses, confirm that the following items are present in the figure legend, table legend, main text, or Methods section.

| n/a | Confirmed | |
|---|---|---|
| ☐ | ☒ | The exact sample size (*n*) for each experimental group/condition, given as a discrete number and unit of measurement |
| ☐ | ☒ | A statement on whether measurements were taken from distinct samples or whether the same sample was measured repeatedly |
| ☐ | ☒ | The statistical test(s) used AND whether they are one- or two-sided *Only common tests should be described solely by name; describe more complex techniques in the Methods section.* |
| ☒ | ☐ | A description of all covariates tested |
| ☒ | ☐ | A description of any assumptions or corrections, such as tests of normality and adjustment for multiple comparisons |
| ☐ | ☒ | A full description of the statistical parameters including central tendency (e.g. means) or other basic estimates (e.g. regression coefficient) AND variation (e.g. standard deviation) or associated estimates of uncertainty (e.g. confidence intervals) |
| ☐ | ☒ | For null hypothesis testing, the test statistic (e.g. *F*, *t*, *r*) with confidence intervals, effect sizes, degrees of freedom and *P* value noted *Give P values as exact values whenever suitable.* |
| ☒ | ☐ | For Bayesian analysis, information on the choice of priors and Markov chain Monte Carlo settings |
| ☒ | ☐ | For hierarchical and complex designs, identification of the appropriate level for tests and full reporting of outcomes |
| ☒ | ☐ | Estimates of effect sizes (e.g. Cohen's *d*, Pearson's *r*), indicating how they were calculated |

*Our web collection on statistics for biologists contains articles on many of the points above.*

## Software and code

Policy information about availability of computer code

| Data collection | Gatan DigitalMicrograph and ThermoFisher EPU v2.9 |
|---|---|
| Data analysis | crYOLO v1.7.5, Topaz v0.2.5, MotionCor2, Gctf v1.06, RELION v3.1, cryoSPARC 3.2, UCSF Chimera v1.14, ChimeraX-1.3, COOT v0.9-pre, Phenix v1.19.2, MolProbity web sever, ImageJ v2.0.0, GraphPad Prism v9.2.0, Phyre2 web server, pyem v0.5, PyMOL v2.4.1 |

For manuscripts utilizing custom algorithms or software that are central to the research but not yet described in published literature, software must be made available to editors and reviewers. We strongly encourage code deposition in a community repository (e.g. GitHub). See the Nature Portfolio guidelines for submitting code & software for further information.

## Data

Policy information about availability of data

All manuscripts must include a data availability statement. This statement should provide the following information, where applicable:

- Accession codes, unique identifiers, or web links for publicly available datasets
- A description of any restrictions on data availability
- For clinical datasets or third party data, please ensure that the statement adheres to our policy

Cryo-EM density maps of the CMGE dimer complex has been deposited in the Electron Microscopy Data Bank (EMDB) under the accession number EMD-13988. Cryo-EM density map of the symmetry expanded CMGE monomer has been deposited in the EMDB under the accession number EMD-13978. Atomic coordinates have been deposited in the Protein Data Bank (PDB) with the accession number 7QHS (symmetry expanded CMGE monomer) and 7Z13 (monomer docked into the CMGE dimer map).

# Field-specific reporting

Please select the one below that is the best fit for your research. If you are not sure, read the appropriate sections before making your selection.

☒ Life sciences　　☐ Behavioural & social sciences　　☐ Ecological, evolutionary & environmental sciences

For a reference copy of the document with all sections, see nature.com/documents/nr-reporting-summary-flat.pdf

# Life sciences study design

All studies must disclose on these points even when the disclosure is negative.

| | |
|---|---|
| Sample size | In our negative stain EM experiments, we imaged ATP-dependent CMG assembly, yielding different reaction intermediates. To isolate CMG dimers, we usually collected 100-300 micrographs per condition, these numbers of micrographs were sufficient to eiter allow 2D classificaiton or comparative analysis between MCM mutants.<br><br>To obtain high-resolution structure of the CMGE nucleating origin DNA melting from the mixed population of reaction intermediates in the cryo-EM experiment, ~65.3 K micrographs were collected from two independent grids made from the same CMG assembly reaction. This number of micrographs was sufficient to either allow model building or comparative analysis.<br><br>No statistical methods were used to predetermine sample size. |
| Data exclusions | For our ReconSil experiments, samples were prepared with reduced concentrations to limit particle crowding and allow the clear identification of single origins of replication. Particles were picked and multiple rounds of 2D classification were performed to isolate particles contributing to the distinct molecular species in our samples. Picked particles that could not be aligned and classified were discarded and therefore were not reconstituted in silico. ReconSiled origins were evaluated and rejected if confident assignment of co-localisation to the same origin could not be made because either, i. the origin was in a region of clustered/aggregated particles or ii. If the origin contained additional particles that had not been 2D classified, and were therefore not overlaid with a 2D class average that would permit confident assignment of the molecular species.<br><br>Negative stain and cryo-EM micrographs with poor staining, ice contamination or entirely lacey carbon, respectively, were excluded. Picked particles that did not align to a distinct class in 2D and 3D (cryo-EM only) were excluded from further analysis. CMGs that were not engaged with pol epsilon were removed from the cryo-EM dataset to yield the best reconstruction of the CMGE dimer complex. |
| Replication | The cryo-EM dataset of ATP-dependent CMG assembly reaction comprised of a single reaction and two datasets, collected on two independent grids. CMG dimer complex formation in negative stain EM experiments was found to be reproducible across multiple independent sample preparations using different DNA substrates. Details of the number of experimental repeats have been acknowledged in the relevant figure legends. All attempts at data replication were successful. Details of the number of experimental repeats have been acknowledged in the relevant figure legends. All attempts at data replication were successful. |
| Randomization | For calculation of the resolution of the cryo-EM reconstructions, Fourier shell correlations were calculated using independent halves of the complete datasets, into which the component particles were segregated randomly. |
| Blinding | Blinding is not relevant for a single particle electron microscopy study such as this. |

# Reporting for specific materials, systems and methods

We require information from authors about some types of materials, experimental systems and methods used in many studies. Here, indicate whether each material, system or method listed is relevant to your study. If you are not sure if a list item applies to your research, read the appropriate section before selecting a response.

## Materials & experimental systems

| n/a | Involved in the study |
|---|---|
| ☒ | ☐ Antibodies |
| ☐ | ☒ Eukaryotic cell lines |
| ☒ | ☐ Palaeontology and archaeology |
| ☒ | ☐ Animals and other organisms |
| ☒ | ☐ Human research participants |
| ☒ | ☐ Clinical data |
| ☒ | ☐ Dual use research of concern |

## Methods

| n/a | Involved in the study |
|---|---|
| ☒ | ☐ ChIP-seq |
| ☒ | ☐ Flow cytometry |
| ☒ | ☐ MRI-based neuroimaging |

# Eukaryotic cell lines

Policy information about cell lines

| | |
|---|---|
| Cell line source(s) | S. cerevisiae overexpression strains for CMG assembly and DNA replication proteins have previously been described in in |

| Cell line source(s) | multiple studies across several publications. For clarity to the potential readers and reviewers we have included extensive details in extended data table 4. |
| --- | --- |
| Authentication | S. cerevisiae overexpression strains were checked for correct plasmid integration by PCR amplification from extracted genomic DNA. |
| Mycoplasma contamination | S. cerevisiae overexpression strains were not tested for mycoplasma contamination. |
| Commonly misidentified lines (See ICLAC register) | No commonly misidentified cell lines were used in this study. |

