## [Peer Review File · Nature]

Manuscript Title: Mechanism of replication origin melting nucleated by CMG helicase assembly

Reviewer Comments & Author Rebuttals

Reviewer Reports on the Initial Version:

Referees' comments:

Referee #1:

Overview: In the field of DNA replication, one long-standing mystery concerns the mechanistic step that initiates the remodelling of the parental DNA duplex into two single-stranded DNA templates (origin melting). The current paper by Lewis et al. provides a detailed and elegant cryo-EM analysis of the origin DNA caught in the act of unwinding in the company of a head-to-head dimeric complex containing the CMG replicative helicase and the leading-strand DNA polymerase epsilon (the dCMGE complex). Somewhat unexpectedly, this novel structure indicates that initial strand separation occurs at separate sites within each of the component helicases rather than in the intervening duplex DNA between them. This structural analysis also provides compelling clues about the coupling between ATP binding and hydrolysis and the conformational changes in the CMG complex that lead to DNA melting and subsequent expulsion of the lagging strand from the complex. Aside from the relatively narrow experimental and presentation issues discussed below, this otherwise well-written manuscript presents compelling data that substantially illuminates this critical but poorly understood replication step.

Problems: However, despite reviewer enthusiasm, there are a few issues:

1. Lack of C-terminal regions in the model?: The summary section of the respective PDB reports that accompany this manuscript indicate that the structural models presented are qualitatively as good as or considerably better than typical cryoEM structures or even crystallographic structures present in the PDB. However, the PDB reports also note that most of the C-terminal portions of key proteins in the complex are excluded from the 3.3 angstrom monomer structure. This “feature” of the model is not discussed in the text or supplemental data. Given that the C-terminal regions of the Mcm subunits contain particularly important elements (e.g., the ATP binding/hydrolysis domains), this seems a bit problematic. I would suppose that these structural limitations may be the root of some

of the below presentation issues regarding the coupling between ATPase activity and DNA unwinding. Please address this issue and as appropriate add a paragraph in the supplemental to both soothe reviewer concerns and explain the structural limitations of the model.

2. “Red Herrings” in the data: Given the manuscript title, I was keen to learn how Mcm2-7 melts the origin DNA. However, the key data (involvement of Mcm2 and Mcm6 in DNA unwinding) is nearly the last thing presented in the Results section and is poorly emphasized relative to earlier presented and lesser results that superficially suggest possible unwinding mechanisms that turn out to be wrong (discussed below). I believe that part of this problem is that only changes to protein structure are emphasized, with short shrift given to conformational changes in the DNA (resolution limitations?). As such, it is difficult for the reader to assemble a clear picture of exactly how the proposed unwinding occurs. Such potential “red herrings” include:

A. The subunit register changes between the double hexamer complex and the dCMGE complex by one subunit (i.e., a 60 degree change in orientation between the two monomer complexes). Does this suggest a 60 degree unwinding of the duplex DNA upon dCMGE formation?

B. Relative to the double hexamer complex, the intervening region between the two CMGE complexes is open and splayed out while pivoting on the Mcm6 connections between the two complexes. Is the intervening DNA stretched in this conformation, and does this potential stress contribute to DNA melting?

An implicit assumption in the above scenarios is that both the double hexamer and the dCMGE complex bind origin DNA in approximately the same location, maintain these DNA contacts during the change in conformation between the double hexamer and the dCME, and thus increase tension in the intervening DNA. The relevant issue only mentioned in passing early in the manuscript and maybe unknown to the naive reader is that the double hexamer can freely slide along double-stranded DNA, strongly suggesting that the complex does not make strong contacts with the DNA. I believe that this observation largely disqualifies the likelihood that the above observations contribute to DNA unwinding, but I may be wrong. Do either of these structural features contribute to DNA unwinding? Briefly address this issue in the text to eliminate potential confusion.

3. The coupling between DNA unwinding and Mcm ATP binding and hydrolysis: Presumably, DNA unwinding is coupled to the ATPase activity of some fraction of the six Mcm active sites, and a section of the results is dedicated to a description of the ATP/ADP occupancy of these six sites in the dCMGE complex. However, it is not especially clear from the results exactly how the changes in the ATPase active site are likely coupled to DNA unwinding. If the data is clear, please directly state; if the data is unclear, this is a good topic for the Discussion section (more below).

4. Poorly articulated DNA unwinding model: As I glean from the paper, origin melting requires two features of the CMGE monomer: 1) The MCM6 wedge provides contacts that stabilize the three basepairs that are unwound in each monomer, and 2) that conformational changes in Mcm2 during dCMG formation are responsible for the actual DNA unwinding. These results are largely presented in the context of the CMGE monomer; however upon reflection, it is clear that a CMGE monomer is not competent to unwind the origin: both complexes in the dimer structure need to act concurrently

to unwind the DNA.

DNA unwinding (as opposed to strand separation) ultimately requires rotation of the Crick strand relative to the Watson strand. As cited in the text, the DNA within the dCMG is underwound by 0.7 turns, or about 250 degrees relative to relaxed duplex DNA. Such unwinding is generally thought to require at least two physical contact points between the DNA and the agent doing the unwinding, similar to two hands twisting in opposite directions to squeeze water out of a soaked bath towel in the region between the two hands; one hand alone cannot twist the water out.

Yet the “one-handed model” is shown in Suppl. Movie 5 and Fig. 5 (again, a resolution concern regarding the dimer structure?). In the movie, the Mcm2 contacts are shown unwinding the neighboring duplex DNA. However, this unwinding cannot actually occur unless the distal end of the DNA is anchored (e.g., by the other CMGE monomer).

Moreover, the presented “one-handed” model specifically shows the three specific basepairs unwound in isolation. As nearly as this reviewer can tell, the Mcm2 conformational changes in the dimer structure just generally decrease the superhelical DNA density between the two complexes, allowing the entire intervening region in the CMGE dimer to “breathe” (transiently unwind). The actual choice of the three unpaired DNA bases is determined by the Mcm6 contacts, which specifically stabilize the unpairing of these three bases relative to the other bases in this region. Clean up movie 5 to include both halves of the dCMG complex, and make clear the distinct and likely sequential contributions that Mcm2 and Mcm6 are making to DNA unwinding. I would suggest that part of the movie should show the change in DNA unwinding in the absence of the proteins to emphasize the DNA structural changes that result in unwinding and melting.

5. The Discussion section: Oddly and contrary to the title of the manuscript, the current Discussion section explores a collection of disparate issues with little direct discussion of how origin melting occurs. It seems prudent to refocus on the underlying problem and use this section instead to provide a clearly articulated model of origin melting that will better sell your hard work. I appreciate that possibly due to potential structure resolution limitations, you are being rather careful in making mechanistic claims in the Results section. However, you can state in the Discussion your interpretation of the structure, and succinctly describe how the data suggest a particular DNA unwinding model. Your model will likely have possible limitations and caveats that you can explore. This is not a bad place to include a low resolution (cartoon) model figure/movie, even if it just gets delegated to Supplemental Data. In short, both structural experts and novices need a clear take-away from this paper, a goal that is not well addressed in the current manuscript.

Lesser issues:

6. There are five movies included as supplemental data. In the text they are listed as 1 through 5. However, two movie files are named “1”, two are named “5”, and a single movie is named “4”. Improved labelling is in order.

7. Figure legend improvements: Several idiosyncratic issues occur in the data that are difficult for the reader to interpret if they are not well versed in papers from the Diffley lab. In most cases this

problem can easily be corrected in the figure legends.

Fig. 1d: In the negative control lanes, inform the reader that the unexpected ~90 kD band is a covalent complex containing the methylase and the DNA substrate.

Fig. 2b: This figure contains either little gray or black dots that separate the subunits of the opposed Mcm structures. Briefly explain what they represent (zinc fingers?).

Fig. 4d: DNA topology assays. There are several oddities about this figure that require elaboration.

1) Ghost bands: It is mentioned in the text that in the absence of Mcm10 DNA unwinding still occurs (“...whilst omission of Mcm10 captures the initial untwisted state. This initially untwisted state generates topoisomers of -2, -3 and -4 (cyan arrow heads)”). The relevant lane in this figure is #2, which does not seem to have any additional underwound bands. Only upon enlarging the image and extremely careful examination can a few faint bands be seen. Add a better gel or a longer exposure.

2) Order of DNA topoisomers in the gel: To the right of Fig. 4d, the DNA topoisomers are numbered. Normally, each topoisomer will show a progressive incremental change in gel mobility, and this forms an numerically ordered array, in this case from unwound (toward the top), to being progressively unwound (toward the bottom). However, in this DNA substrate, contrary to viewer expectations, the numbering is not sequential. Apparently, this is a known issue with this particular DNA substrate, although the exact reason for such weirdness was not conclusively shown in the cited reference. To relieve reader angst, briefly mention the relevant information in the legend.

Referee #2:

In this manuscript, the authors have once again utilized cryo-EM to capture a unique state of CMG loading onto chromatinized DNA template. They find that the two CMG hexamers will form a splayed dimer that acts to untwist and begin to break base pairing within the central channels of the hexamers. This conformation change and state is introduced by ATP-dependent sequential binding throughout the hexamer. Interestingly, this conformation also shifts in the register for the interacting MCM subunits (2/6 and 5/7) that removes a helical contact from MCM7 to MCM5 that then allows GINS binding to further activate the hexamers. This work is extremely well done and provides significant further insights into the CMGE activation mechanism. The discussion does a great job stepping the reader through the implications of the new results, including a mechanism for controlling E3 ligase binding.

1) You should describe the Mcm2 6A mutant structurally in a supplemental panel, highlighting the residues that are mutated and their interactions with either ATPase or DNA. I found this hard to find.

2) Can you discuss whether both CMGE complexes are expected to shift together to cause the register shift or whether one CMGE hexamer preferentially provides this realignment?

3) Lines 220 and 235: Do the broken base-pairing residues on the leading strand form H-bonds with the MCM6 residues, T423 and R424, or with those other identified residues from MCM2 on the lagging strand? In both cases you mention contacts and stabilisation without being specific. Also

what are those DNA base pairs that are broken? Preferentially A-T, or does it not matter? Maybe a supplemental figure showing more atomic structures and contacts of these interactions could be useful.

Referee #3:

This study by Lewis et al. reports the cryo-EM structure of dimeric *S. cerevisiae* replicative helicases (CMGs) on path towards full activation during origin firing. Helicase activation is a key step during replication initiation, and how replicative helicases are activated and how bidirectional replication forks are established have been long-standing questions. The authors' work provides the first structural snapshot of how these steps are accomplished. In comparing the current dimeric CMGE structure to that of the MCM double hexamer previously determined, the authors observe large conformational changes in the MCM rings as a result of nucleotide exchange at the ATPase sites, which leads to rearrangements in the MCM dimerization interface, DNA untwisting, and duplex melting. This work is of high quality and the authors convincingly support their structural observations biochemically through mutational analyses of key elements, such as the Mcm2 pore loop and the Mcm6 wedge. In conclusion, this structure represents a landmark not only for the replication field but will undoubtedly be of interest to multiple disciplines and the broad readership of Nature.

The following points should be considered by the authors:

1. The observation of cis-dCMGEs is very interesting. Are they caused by rotation of the CMGEs with respect to each other after dimer separation or do they result from a different type of MCM double hexamer that is loaded onto DNA?
2. Including the mutant MCM data in Fig. 1d and 1f is confusing since these data are not discussed until later in the manuscript and are also not described in the figure legend. It would be advisable to move these data to a later figure, for example Fig. 4. In addition, the authors should explain what the strong band corresponds to around 100 kDa that elutes in all reactions in 1d.
3. Ext. Data Fig. 1: Considering the heterogeneity in the sample, it would be helpful to circle the particles in the micrographs that correspond to the class averages shown below in 1c and 1f. For panel 1g, it is not clear what is shown in lane 1.
4. The cartoon in Fig. 2b for the subunit transitions between MCM-DH and dCMGE is somewhat confusing. It would be helpful to adjust the subunit positions of Mcm2 and 6 in the cartoon to their position in the structure shown. For example, in dCMGE the left Mcm6 is shown below the right Mcm6, yet in the structure it is opposite. This also applies to the corresponding movie, where the moving parts (MCM hexamer) in the structure and schematic do not correspond to each other.
5. In Fig. 2e, the authors should comment on why pol e is not visible in CMG class averages after MseI addition to dCMGE.

6. In Fig. 3c, which structure is shown on the right of this panel? If it is the same as on the left?

7. In Ext. Data Fig. 5, it is quite difficult to judge in panels c and d how well the model fits the map density. The authors may want to choose a different rendering style for the cryo-EM map. W589 in Mcm2 is not shown in panel a although it binds DNA and should be included.

Author Rebuttals to Initial Comments:

Response to reviewers' comments

We would like to thank all referees for the positive reviews and useful comments.

Referee #1:

We are grateful to the reviewer for expressing their “enthusiasm” and stating that our cryo-EM analysis is “detailed and elegant”. It is pleasing to read that the paper is deemed “well-written” and the data “compelling” and “substantially illuminat[ing] this critical ... replication step”.

1. Lack of C-terminal regions in the model?: The summary section of the respective PDB reports that accompany this manuscript indicate that the structural models presented are qualitatively as good as or considerably better than typical cryoEM structures or even crystallographic structures present in the PDB.

We would like to thank this reviewer for noting the quality of the structural model, which is described in the PDB report.

However, the PDB reports also note that most of the C-terminal portions of key proteins in the complex are excluded from the 3.3 angstrom monomer structure. This “feature” of the model is not discussed in the text or supplemental data. Given that the C-terminal regions of the Mcm subunits contain particularly important elements (e.g., the ATP binding/hydrolysis domains), this seems a bit problematic. I would suppose that these structural limitations may be the root of some of the below presentation issues regarding the coupling between ATPase activity and DNA unwinding. Please address this issue and as appropriate add a paragraph in the supplemental to both soothe reviewer concerns and explain the structural limitations of the model.

We respectfully point out that the reviewer is incorrect when stating that most C-terminal portions of the protein chains in our structure are missing. [Redacted]

The grey segment, representing the fraction of residues that are not modelled, is always reported on the right of the bar chart but this does not represent the C-terminal region of the chain. Likewise, the green bar on the left represents the fraction of the chain containing 0 geometric quality outliers, but this does

not mean that it corresponds to the N-terminal region of the chain. Indeed, most Mcm proteins contain unstructured N-terminal tails that have not been built in any of the structures released in the Protein Data Bank and are not built in our new structure.

The table below summarises the geometric issues observed across the polymeric chains and their fit to the map. The red, orange, yellow and green segments of the bar indicate the fraction of residues that contain outliers for ≥ 3 , 2, 1 and 0 types of geometric quality criteria respectively. A grey segment represents the fraction of residues that are not modelled. The numeric value for each fraction is indicated below the corresponding segment, with a dot representing fractions $\leq 5\%$. The upper red bar (where present) indicates the fraction of residues that have poor fit to the EM map (all-atom inclusion $< 40\%$). The numeric value is given above the bar.

Mol	Chain	Length	Quality of chain
1	2	868	2	3	1006	3	4	933	4	6	1017	5	7	845	6	H	208	7	I	213	8	C	229	
Another useful example is the Pol2 chain, of which we built the C-terminal but not the N-terminal domain.

The flexibly tethered (invisible) N-terminal domain, which is absent in our atomic model, is accounted for with the grey bar on the right and not on the left.

We are confident that this explanation will soothe the reviewer. [Redacted] The PDB reports file was submitted as material to be assessed by the reviewers and is not part of the supplementary information intended for publication.

2. “Red Herrings” in the data: Given the manuscript title, I was keen to learn how Mcm2-7 melts the origin DNA. However, the key data (involvement of Mcm2 and Mcm6 in DNA unwinding) is nearly the last thing presented in the Results section and is poorly emphasized relative to earlier presented and lesser results that superficially suggest possible unwinding mechanisms that turn out to be wrong

(discussed below). I believe that part of this problem is that only changes to protein structure are emphasized, with short shrift given to conformational changes in the DNA (resolution limitations?).

We agree with this reviewer. In our original manuscript, we should have done a better job at explaining how a change in inter-ring interactions within the CMGE dimer relates to DNA unwinding. We followed the reviewer's advice and added a sentence in the Results to inform the reader that the register shift in ring dimerisation has implications for origin DNA unwinding. We now state:

“transition to dCMGE involves loss of several trans-ring interactions and a 1-subunit register shift for the remaining tethering elements. What role this register shift might play during origin unwinding will be addressed in the Supplementary Discussion section” .

The reviewer brings up the important issue of limitations in resolution for the reconstruction of the CMGE dimer. We would like to clarify that we employed symmetry expansion as implemented in Relion (Scheres et al. 2016, PMID: 27572726), to be able to describe each of the two rings in our dimer structure, to 3.4 Å resolution. This procedure is required because the tethering elements that connect the CMGE dimer are tenuous, resulting in a flexible dimer, whose two-fold symmetric character is imperfect. Symmetry expansion procedures reveal that the structure of the two rings is identical. This means that two copies of the single ring reconstruction can be overlaid to the lower resolution dimer map, to obtain a reliable 3.4 Å average resolution model of the CMGE dimer. This procedure allows us to confidently state that the duplex DNA mapping between the two CMGE complexes is not untwisted.

To clarify this key issue we changed the Results section, which now reads:

“However, symmetry expansion approaches revealed that two rings in the dimer are identical. Combined with 3D classification, variability analysis, and refinement, followed by iterative cycles of CTF refinement and Bayesian polishing, this process yielded a 3.5 Å resolution structure²⁵⁻²⁷ (or 3.4 Å after density modification²⁸, **Fig. 2a, Supplementary Video 1, Extended Data Fig. 2g–j, 3, and 4, Extended Data Table 1**). By overlaying two copies of the CMGE monomer to the lower resolution dimer, we can therefore obtain a high-resolution view of the entire dCMGE assembly.”

As such, it is difficult for the reader to assemble a clear picture of exactly how the proposed unwinding occurs. Such potential “red herrings” include:

A. The subunit register changes between the double hexamer complex and the dCMGE complex by one subunit (i.e., a 60 degree change in orientation between the two monomer complexes). Does this suggest a 60 degree unwinding of the duplex DNA upon dCMGE formation?

DNA untwisting in the ATPase domain is related to the register shift within the MCM dimerisation interface. The reviewer's proposition that a 60 degree ring rotation would result in 60 degree unwinding however would only be applicable if MCM monomers transition from a DH to dCMGE as rigid bodies. This is not the case. In fact, within individual polypeptide chains, ATPase domains involved in DNA melting move relative to the N-terminal domains involved in inter-ring contacts. For this reason, MCM rings rotate relative to one another less than what the leading strand rotates around the lagging strand. Although this explanation addresses a sharp point raised by the reviewer, we feel that adding this remark to the Results section would be distracting to the reader.

B. Relative to the double hexamer complex, the intervening region between the two CMGE complexes is open and splayed out while pivoting on the Mcm6 connections between the two complexes. Is the intervening DNA stretched in this conformation, and does this potential stress contribute to DNA melting?

No significant stretching/underwinding can be observed in the intervening duplex DNA that connects two MCM rings in the dCMGE. This is because the N-terminal collar rotates around duplex DNA at the MCM dimerisation interface, while no protein–DNA interaction between N-terminal MCM and DNA is maintained upon transition between the DH and dCMGE complexes. This is an important point, which is now addressed in the Supplementary Discussion. In order for the intervening duplex DNA to become unwound on the path to replication fork establishment, it is conceivable that N-terminal MCM–DNA contacts established upon dCMGE formation (*e.g.* the Mcm6 wedge) are maintained as the helicase transitions to downstream intermediates towards duplex DNA unwinding.

An implicit assumption in the above scenarios is that both the double hexamer and the dCMGE complex bind origin DNA in approximately the same location, maintain these DNA contacts during the change in conformation between the double hexamer and the dCME, and thus increase tension in the intervening DNA. The relevant issue only mentioned in passing early in the manuscript and maybe unknown to the naive reader is that the double hexamer can freely slide along double-stranded DNA, strongly suggesting that the complex does not make strong contacts with the DNA. I believe that this observation largely disqualifies the likelihood that the above observations contribute to DNA unwinding, but I may be wrong. Do either of these structural features contribute to DNA unwinding? Briefly address this issue in the text to eliminate potential confusion.

When comparing DH and dCMGE, we observe that certain C-terminal ATPase elements (*e.g.* Mcm2 K587) maintain the same DNA contact. This is instead not the case for N-terminal collar elements that form the MCM dimerisation interface. For this reason, DNA melting only occurs within the ATPase tier of MCM, while the N-terminal tier only rotates around the duplex DNA stretch that becomes exposed between the two rings. The result of this rotation is a register shift in the MCM ring dimer upon transition from DH to dCMGE. Such rearrangement diffuses topological strain outside of the dimer, rather than focusing it towards duplex DNA intervening between the two rings. This is now addressed in the Supplementary Discussion and in Extended Data Figure 8.

3. The coupling between DNA unwinding and Mcm ATP binding and hydrolysis: Presumably, DNA unwinding is coupled to the ATPase activity of some fraction of the six Mcm active sites, and a section of the results is dedicated to a description of the ATP/ADP occupancy of these six sites in the dCMGE complex.

However, it is not especially clear from the results exactly how the changes in the ATPase active site are likely coupled to DNA unwinding. If the data is clear, please directly state; if the data is unclear, this is a good topic for the Discussion section (more below).

We agree with this reviewer that some context should be given in the ATPase paragraph for how nucleotide occupancy affects DNA unwinding. In the Results section, we now state:

“This nucleotide exchange in turn alters the way duplex DNA is gripped by ATPase pore loops of the MCM.”

In addition, we conclude the nucleotide occupancy paragraph by stating **“How nucleotide-triggered conformational changes in MCM affect duplex DNA structure will be discussed in the next paragraphs.”** This refers to the Results section “Mechanism of DNA-bubble nucleation”, where the ATPase-controlled reconfiguration of the Mcm2 elements PS1BH and h2i is discussed. The new sentence also refers to the section “Open DNA stabilised as MCM dimer splits” where the ATPase-controlled reconfiguration of Mcm4 and Mcm7 pore loops creates the space for MCM pore invasion by the Mcm6 wedge.

4. Poorly articulated DNA unwinding model:

As I glean from the paper, origin melting requires two features of the CMGE monomer: 1) The MCM6 wedge provides contacts that stabilize the three basepairs that are unwound in each monomer, and 2) that conformational changes in Mcm2 during dCMG formation are responsible for the actual DNA unwinding. These results are largely presented in the context of the CMGE monomer; however upon reflection, it is clear that a CMGE monomer is not competent to unwind the origin: both complexes in the dimer structure need to act concurrently to unwind the DNA.

This event requires an MCM dimer and its conversion to two CMGEs.

What the reviewer states about *origin unwinding* is correct, however this is distinct from the nucleation of origin melting, which we describe in our study. Full origin unwinding involves lagging strand

ejection from the MCM ring pore and the bypass of two helicases to establish two diverging replication forks. These events are triggered as a result of Mcm10 recruitment. In a 2019 eLife paper (PMID: 31282859), Langston and O'Donnell show that only two converging CMG helicases translocating along duplex DNA can unwind the double helix, with lagging strand ejection from the MCM ring pore promoted by Mcm10. One single CMG cannot do the job.

In a different 2019 eLife paper (PMID: 31385807), Champasa, Gelles, Bell, and colleagues demonstrate that the loading of an MCM double hexamer but not a single hexamer supports full origin unwinding. In fact, a MCM variant, which can be loaded onto duplex DNA as a monomer, but cannot dimerise, fails to extensively unwind DNA when firing factors are added. However, the authors also show that a single MCM loaded onto origin DNA supports CMG formation and DNA untwisting to the same extent as wild type CMG assayed in the absence of Mcm10. CMG formation in the absence of Mcm10 is precisely what we describe in the present study, here referred to as 'nucleation of origin DNA melting'. Thus, we feel that our focus on DNA melting inside one single MCM ring extracted from the dCMGE structure is appropriate.

We now cite the Champasa and the Langston articles in the Supplementary Discussion, where we also discuss a possible mechanism for full origin DNA unwinding.

DNA unwinding (as opposed to strand separation) ultimately requires rotation of the Crick strand relative to the Watson strand. As cited in the text, the DNA within the dCMG is underwound by 0.7 turns, or about 250 degrees relative to relaxed duplex DNA. Such unwinding is generally thought to require at least two physical contact points between the DNA and the agent doing the unwinding, similar to two hands twisting in opposite directions to squeeze water out of a soaked bath towel in the region between the two hands; one hand alone cannot twist the water out.

The reviewer brings up an important point. In order to nucleate DNA melting, at least two tethering elements ("two hands") must engage DNA. The question is whether these two elements exist within one single CMGE complex or within the dimer. We would like to point out that in the Results paragraph entitled "Mechanism of DNA-bubble nucleation" we indeed describe separate elements that touch either strand in the DNA. One (Mcm2 K587) pushes the lagging strand template. The other element (Mcm2 V580, K582, P584, W589 and K633) pull on the leading strand template. As a result, the two strands are pulled apart and the minor groove is widened. This event occurs within one single dCMGE complex extracted from the symmetric dimer. We refer to Supplementary Video 4 for a depiction of this structural transition.

This observation should have been presented better in the first version of our manuscript. We improved the description of the DH-to-dCMGE structural transitions in the Results section by now stating (key change highlighted in *italic*):

“Structural changes in the ATPase pore loops explain how DH-to-dCMGE transition leads to DNA untwisting. Amongst several MCM–DNA interactions that are summarised in **Extended Data Fig. 5a**, we identified K587 on the Mcm2 helix-2 insert (h2i) pore loop as one of few elements that maintain the same DNA contact in both the DH and dCMGE. *As shown in Fig. 4c and Supplementary Video 4, this element appears to push on the lagging-strand template, contributing to duplex DNA deformation. Additional DNA contacts involve five conserved residues on Mcm2 that pull on the leading-strand template (V580, K582, P584, W589 in h2i, as well as K633 in the pre-sensor1 beta hairpin, PS1BH, Extended Data Figure 5f).*”

Yet the “one-handed model” is shown in Suppl. Movie 5 and Fig. 5 (again, a resolution concern regarding the dimer structure?). In the movie, the Mcm2 contacts are shown unwinding the neighboring duplex DNA. However, this unwinding cannot actually occur unless the distal end of the DNA is anchored (e.g., by the other CMGE monomer).

We summarised above biochemical data from the Bell lab (PMID: 31385807) indicating that DNA untwisting upon CMG formation at origins does not require a dimer of MCMs. Thus, a second anchoring element that leads to untwisting/melting nucleation is unlikely to be the second MCM ring. Extended Data Figure 5a shows several ATPase-DNA contacts, other than Mcm2, which likely contribute to origin untwisting and bubble nucleation.

Taken together, our structural observations, combined with the biochemical evidence from the Bell lab that a single CMG can untwist DNA, indicate that it is appropriate for us to discuss DNA melting nucleation in the context of one single ring.

Moreover, the presented “one-handed” model specifically shows the three specific basepairs unwound in isolation. As nearly as this reviewer can tell, the Mcm2 conformational changes in the dimer structure just generally decrease the superhelical DNA density between the two complexes, allowing the entire intervening region in the CMGE dimer to “breathe” (transiently unwind).

The actual choice of the three unpaired DNA bases is determined by the Mcm6 contacts, which specifically stabilize the unpairing of these three bases relative to the other bases in this region.

The reviewer is correct when stating that conformational changes in Mcm2 decrease superhelical DNA density. We should point out that this effect is localised to a stretch of 7 base pairs, as observed in our structure, and does not extend to the entire intervening region between two Mcm2 protomers in the dCMGE. The amount of underwinding observed in our structure accounts for the entirety of DNA untwisting previously observed in topology footprint experiments (Douglas et al Nature 2018 PMID: 29489749). Thus, modelling further unwinding in the intervening region would be incompatible with biochemical evidence.

Following the reviewer's suggestion, we now state in the results: "These [Mcm2] contacts are found in the dCMGE but are absent in the DH and widen the minor groove, decreasing superhelical DNA density (Fig. 4c)."

The reviewer is also correct when stating that 3 bases can be nicely resolved because they are stabilised by the Mcm6 wedge. In the Results section we state (key passage highlighted in *italic*):

"When comparing the DH and dCMGE structures we observed that the Mcm6 wedge insertion, *which stabilises the lagging strand orphan bases in the dCMGE*, is retracted and contributes to stabilising the dimerisation interface in the DH (Fig. 5a).

Clean up movie 5 to include both halves of the dCMG complex,

Supplementary Video 5 now starts with a focus on the MCM double hexamer interface, then shows the structural changes within one MCM hexamer and ends with displaying the structural transition in the DNA occurring in the context of the dimeric structure.

and make clear the distinct and likely sequential contributions that Mcm2 and Mcm6 are making to DNA unwinding.

We have changed Supplementary Video 5 to specify that the Mcm6 wedge stabilises three orphan bases in the lagging-strand template.

I would suggest that part of the movie should show the change in DNA unwinding in the absence of the proteins to emphasize the DNA structural changes that result in unwinding and melting.

We thank the reviewer for their suggestion and have now implemented this change to Supplementary Video 5.

5. The Discussion section: Oddly and contrary to the title of the manuscript, the current Discussion section explores a collection of disparate issues with little direct discussion of how origin melting occurs.

It seems prudent to refocus on the underlying problem and use this section instead to provide a clearly articulated model of origin melting that will better sell your hard work. I appreciate that possibly due to potential structure resolution limitations, you are being rather careful in making mechanistic claims in the Results section. However, you can state in the Discussion your interpretation of the structure, and succinctly describe how the data suggest a particular DNA unwinding model. Your model will likely have possible limitations and caveats that you can explore.

Encouraged by the reviewer we now use Extended Data Fig. 8 to present a more exhaustive model for origin unwinding. This is accompanied by additional text in the Supplementary Discussion.

This is not a bad place to include a low resolution (cartoon) model figure/movie, even if it just gets delegated to Supplemental Data. In short, both structural experts and novices need a clear take-away from this paper, a goal that is not well addressed in the current manuscript.

As requested, we added a cartoon depicting origin unwinding in the Extended Data Fig. 8. (See below).

Lesser issues:

6. There are five movies included as supplemental data. In the text they are listed as 1 through 5. However, two movie files are named “1”, two are named “5”, and a single movie is named “4”. Improved labelling is in order.

We have now corrected the labelling. Thank you for bringing this issue to our attention.

7. Figure legend improvements: Several idiosyncratic issues occur in the data that are difficult for the reader to interpret if they are not well versed in papers from the Diffley lab. In most cases this problem can easily be corrected in the figure legends.

We have now added a diagram in a new Extended Data Fig. 6 that depicts the topology experiment and use the figure legend to better describe the assay.

Fig. 1d: In the negative control lanes, inform the reader that the unexpected ~90 kD band is a covalent complex containing the methylase and the DNA substrate.

Thank you for pointing this out. We have now included an arrowhead that points at the ~90 kDa band and state in the figure legend “Black arrowhead indicates MH bound DNA.”

Fig. 2b: This figure contains either little gray or black dots that separate the subunits of the opposed Mcm structures. Briefly explain what they represent (zinc fingers?).

The reviewer is right in stating that the dots should be explained. The black dots represent zinc fingers engaged in tight inter-ring interactions. The legend of Fig. 2b now reads: “Circles represent zinc fingers. Black circles connected by lines indicate zinc fingers engaged in tight inter-ring interactions.”

Fig. 4d: DNA topology assays. There are several oddities about this figure that require elaboration. 1) Ghost bands: It is mentioned in the text that in the absence of Mcm10 DNA unwinding still occurs (“...whilst omission of Mcm10 captures the initial untwisted state. This initially untwisted state generates topoisomers of -2, -3 and -4 (cyan arrow heads”). The relevant lane in this figure is #2, which does not seem to have any additional underwound bands. Only upon enlarging the image and extremely careful examination can a few faint bands be seen. Add a better gel or a longer exposure.

Thank you for bringing up an important point. In the new Extended Data Figure 6c we now include an over-exposed image of the DNA topology assay gel, compared to what is included in Fig. 4d. With the new exposure, topoisomers -2 and -3 are now clearly visible.

2) Order of DNA topoisomers in the gel: To the right of Fig. 4d, the DNA topoisomers are numbered. Normally, each topoisomer will show a progressive incremental change in gel mobility, and this forms an numerically ordered array, in this case from unwound (toward the top), to being progressively unwound (toward the bottom). However, in this DNA substrate, contrary to viewer expectations, the numbering is not sequential. Apparently, this is a known issue with this particular DNA substrate, although the exact reason for such weirdness was not conclusively shown in the cited reference. To relieve reader angst, briefly mention the relevant information in the legend.

The reviewer is correct in pointing out the non-linear relationship between increased negative supercoiling and electrophoretic mobility (for example, compare mobility of topoisomers α -3 and α -4). This is a well-documented phenomenon and has been observed previously (Zivanovic et al J Mol Biol 1986 PMID: 3560230) and may reflect extrusion of cruciform DNA, which is favoured as linking number decreases (T.S. Hsieh & J.C.Wang, Biochemistry 1975 PMID: 1111569). We have now changed the figure legend of Figure 4d to cite Douglas et al Nature 2019 (PMID: 29489749), in which the individual topoisomers for this particular DNA substrate have been empirically determined.

The caption now reads (change highlighted in bold):

Topology footprint assay for DNA unwinding. Complete reactions contained all firing factors after MCM loading plus TopoI; omission of DDK blocks all untwisting. Omission of Mcm10 captures the initial untwisted state¹. This initially untwisted state generates topoisomers of -2, -3 (cyan arrow heads) **as previously observed**¹. Additional negatively supercoiled topoisomers can be detected when Mcm10 is present, indicating further untwisting upon lagging strand ejection from CMG. No topoisomers were observed with the Mcm2 6A mutant. For gel source data, see Supplementary Fig. 1.

To better explain this assay, in the new Extended Data Fig. 6b we have included an outline of the DNA topology assay in both written and graphical form to aid readers.

Referee #2:

We thank this reviewer for stating that our work is “extremely well done”, and that “the discussion does a great job”.

1) You should describe the Mcm2 6A mutant structurally in a supplemental panel, highlighting the residues that are mutated and their interactions with either ATPase or DNA. I found this hard to find.

All residues targeted by the Mcm2 6A mutant are labelled in Figure 4d, but we realise that this is a busy panel. In Extended Data Fig. 6a we now added a view of the isolated Mcm2 ATPase domain highlighting the 6 residues targeted by mutagenesis (also shown below).

2) Can you discuss whether both CMGE complexes are expected to shift together to cause the register shift or whether one CMGE hexamer preferentially provides this realignment?

Our effort to reconstruct of the dCMGE without symmetry imposition, as well as our symmetry expansion analysis, both fail to identify significant asymmetric features between the two CMGEs. For this reason, our structure alone provides no element to comment on any sequential step towards achieving a shift in MCM register. Future experiments aimed at visualising structural intermediates towards dCMGE formation will hopefully address this question. We modified the legend to Supplementary Video 2, which now reads: “Whether the DH to dCMGE transition is symmetric or conformational changes occur sequentially within the two rings remains an open question.”

3) Lines 220 and 235: Do the broken base-pairing residues on the leading strand form H-bonds with the MCM6 residues, T423 and R424, or with those other identified residues from MCM2 on the lagging strand? In both cases you mention contacts and stabilisation without being specific.

We do not have the resolution to comment on this issue. Part of the problem has to do with the uncertainty about the DNA sequence assignment in our structure, further addressed below.

Also what are those DNA base pairs that are broken? Preferentially A-T, or does it not matter? Maybe a supplemental figure showing more atomic structures and contacts of these interactions could be useful.

We now address this point in the Model building and refinement paragraph of the Methods section by stating: “The register of origin DNA engagement of dCMGE is heterogeneous because MCM double hexamers can slide along duplex DNA before dCMGE is formed. For this reason we could not build the origin DNA sequence with certainty and modelled polyA:polyT DNA instead.”

Referee #3:

We thank this reviewer for stating that our work “is of high quality” and that our mutational analysis “convincingly support[s]” our structural observations. It is great that this reviewer considers our structure “a landmark”, which will “undoubtedly be of interest to the broad readership of Nature”.

The following points should be considered by the authors:

1. The observation of *cis*-dCMGEs is very interesting. Are they caused by rotation of the CMGEs with respect to each other after dimer separation or do they result from a different type of MCM double hexamer that is loaded onto DNA?

MCM double hexamers have been studied by multiple groups both in their DNA-bound (Abid Ali et al Nat Comm 2017: PMID: 29269875; Noguchi et al. PNAS 2017: PMID: 29078375; Greiwe et al NSMB 2022: PMID: 34963704) and DNA-free forms (Li et al Nature 2015; PMID: 26222030). All reported structures describe only one inter-ring register. For this reason, we infer that the *cis*-dCMGE is likely established after *trans*-dCMGE separation. Prompted by this reviewer’s remark, we decided to further elaborate on the *cis*-dCMGE in our manuscript. We recognise that this point is rather speculative, and we do not know whether such a minor particle population (11% of CMGs) represents a *bona fide* helicase activation intermediate on the path to replication fork establishment. Thus, we decided to add a comment on the *cis*-dCMGE in the caption of Extended Data Fig. 1, and not in the main text. The legend for Extended Data Fig. 1c now reads: “70% of CMG particles exist in a dimeric (dCMGE) *trans* configuration (light orange), with GINS positioned on opposed sides of MCM. 11% of dCMGE particles exist in a *cis* configuration (dark orange) that might derive from *trans* dCMGE disengagement and rotation.”

2. Including the mutant MCM data in Fig. 1d and 1f is confusing since these data are not discussed until later in the manuscript and are also not described in the figure legend. It would be advisable to move these data to a later figure, for example Fig. 4.

We are sorry that this reviewer finds the position of Fig. 1d and 1f confusing. We do need to introduce MCM double hexamer loading and DNA replication with wild type protein at the start of the results section. Showing the mutants in a separate panel would require replicating the wild type experiment, which is not ideal given the tight space constraints that come with a Nature article format. Our solution is not to change the position of Fig. 1d and 1f, but to do a better job at explaining the panels in the figure caption. The legend for Fig. 1f now reads: “Replication reaction conducted as shown in (d) except on large ARS1 circular DNA of wild type and mutant MCMs. Mutants include Mcm2 6A that targets residues involved in DNA untwisting, Mcm6 2E that targets the Mcm6 wedge insertion and Mcm6 5E that targets the safety latch. For gel source data, see Supplementary Fig. 1.”

In addition, the authors should explain what the strong band corresponds to around 100 kDa that elutes in all reactions in 1d.

The band migrating around ~100kDa corresponds to the covalent methyltransferase bound to one denatured strand of the short DNA substrate. We added a black arrow pointing at the 100kDa band and now state in the Fig. 1d caption: “Black arrowhead indicates MH bound DNA.” See also response to reviewer #1 point 7.

3. Ext. Data Fig. 1: Considering the heterogeneity in the sample, it would be helpful to circle the particles in the micrographs that correspond to the class averages shown below in 1c and 1f.

Great suggestion. We have now added colour coded circles to particles and corresponding 2D class averages.

For panel 1g, it is not clear what is shown in lane 1.

Thanks for pointing this out. The caption for Extended data Fig. 1g now reads: “6% PAGE gel of partial DNA digestion of 6x ARS array by *MseI* carried out under the same conditions as NS-EM experiments. Lane 1 contains unmodified 6x ARS1 array. Lane 2 contains MH-conjugated 6x ARS1 array DNA. Lane 3 contains *MseI* digested 6x ARS1 array DNA.”

4. The cartoon in Fig. 2b for the subunit transitions between MCM-DH and dCMGE is somewhat confusing. It would be helpful to adjust the subunit positions of Mcm2 and 6 in the cartoon to their position in the structure shown. For example, in dCMGE the left Mcm6 is shown below the right Mcm6, yet in the structure it is opposite.

We adjusted the cartoons to better reflect the MCM register in the double hexamer and dCMGE structures. Panel 2b now looks like this:

This also applies to the corresponding movie, where the moving parts (MCM hexamer) in the structure and schematic do not correspond to each other.

This is true. The cartoons in Supplementary Video 2 have been adjusted to best reflect the structure. Thank you.

5. In Fig. 2e, the authors should comment on why pol e is not visible in CMG class averages after MseI addition to dCMGE.

This is a good point. We altered the caption to for Fig. 2e, which now reads: “DNA digestion disrupts the dCMGE dimer into single isolated CMGs, while also promoting Pol epsilon disengagement.”

6. In Fig. 3c, which structure is shown on the right of this panel? If it is the same as on the left?

The reviewer is correct. We modified the caption for Fig. 3c, which now reads: “The density for the selected translocation strand (red on the right) has been extracted from the duplex-DNA density (grey on the left).”

7. In Ext. Data Fig. 5, it is quite difficult to judge in panels c and d how well the model fits the map density. The authors may want to choose a different rendering style for the cryo-EM map.

We now include improved rendering styles for Extended Data Fig. 5c and d. Thank you for pointing this out.

W589 in Mcm2 is not shown in panel a although it binds DNA and should be included.

Apologies for the oversight. We now corrected Extended Data Fig. 5a.

Reviewer Reports on the First Revision:

Referees' comments:

Referee #1:

My previous concerns on this manuscript have now been alleviated. Nice revision!

Small matter - spelling issue on the new Ext. Data Fig. 8 – “reconfiguration”.

Referee #3:

The authors have sufficiently addressed the previous concerns and the revised manuscript is suitable for publication.